# NAD$^+$ analog reveals PARP-1 substrate-blocking mechanism and allosteric communication from catalytic center to DNA-binding domains

Marie-France Langelier[1], Levani Zandarashvili[2], Pedro M. Aguiar ID [3], Ben E. Black[2] & John M. Pascal[1]

PARP-1 cleaves NAD$^+$ and transfers the resulting ADP-ribose moiety onto target proteins and onto subsequent polymers of ADP-ribose. An allosteric network connects PARP-1 multi-domain detection of DNA damage to catalytic domain structural changes that relieve catalytic autoinhibition; however, the mechanism of autoinhibition is undefined. Here, we show using the non-hydrolyzable NAD$^+$ analog benzamide adenine dinucleotide (BAD) that PARP-1 autoinhibition results from a selective block on NAD$^+$ binding. Following DNA damage detection, BAD binding to the catalytic domain leads to changes in PARP-1 dynamics at distant DNA-binding surfaces, resulting in increased affinity for DNA damage, and providing direct evidence of reverse allostery. Our findings reveal a two-step mechanism to activate and to then stabilize PARP-1 on a DNA break, indicate that PARP-1 allostery influences persistence on DNA damage, and have important implications for PARP inhibitors that engage the NAD$^+$ binding site.

[1] Department of Biochemistry and Molecular Medicine, Université de Montréal, Montréal, QC H3C 3J7, Canada. [2] Department of Biochemistry and Biophysics, Perelman School of Medicine, University of Pennsylvania, Philadelphia, PA 19104-6059, USA. [3] Department of Chemistry, Université de Montréal, Montréal, QC H3C 3J7, Canada. Correspondence and requests for materials should be addressed to B.E.B. (email: blackbe@pennmedicine.upenn.edu) or to J.M.P. (email: john.pascal@umontreal.ca)

Poly(ADP-ribose) polymerase-1 (PARP-1) is an enzyme that uses $NAD^+$ to produce the posttranslational modification poly(ADP-ribose) (PAR) attached to PARP-1 itself or other target proteins[1]. PARP-1 participates in multiple cellular processes, most notably DNA damage repair, transcriptional regulation, and cell death signaling[2]. In DNA repair, PARP-1 rapidly detects DNA strand break damage and recruits repair factors through the local production of PAR[3]. PARP-1 is the founding member of the PARP superfamily, which includes 17 members with a conserved catalytic region with an ADP-ribosyl transferase (ART) fold, but a distinct array of regulatory domains that dictate their biochemistry and cellular functions[4]. Several PARP family members have emerged as promising therapeutic targets, primarily for cancer treatment, thus underscoring the need to understand the mechanism of action and regulation of PARP enzymes.

PARP-1 has a low level of basal catalytic activity that is highly stimulated up to 1000-fold by DNA strand breaks[5]. PARP-1 binding to DNA strand break damage is achieved through coordinated action of two zinc finger domains, Zn1 and Zn2, located at the N-terminus of the protein (Fig. 1a)[6, 7]. A third zinc-binding domain with an unrelated protein fold, Zn3, and the WGR (Trp-Gly-Arg) domain also interact with DNA[8]. These regulatory domains form mutually compatible contacts with damaged DNA, and this domain assembly on DNA leads to the formation of interdomain contacts that are essential for DNA damage-dependent catalytic activation of PARP-1[8]. The crystal structure of PARP-1 essential domains on a DNA double-strand break indicated a structural transition in the helical subdomain (HD) region of the catalytic domain (CAT) that occurred in response to PARP-1 interaction with DNA[8]. We have recently used hydrogen/deuterium exchange with mass spectrometry (HXMS) to measure changes in PARP-1 dynamics owing to DNA damage detection and found that specific helices within the HD exhibit marked increases in hydrogen exchange, consistent with unfolding of these helices or rapid sampling of the unfolded state when PARP-1 binds to DNA strand breaks[9]. Deletion of the HD produces an overactive enzyme and fully recapitulates the effect of PARP-1 DNA break binding on PAR catalysis, indicating that the HD acts as an autoinhibitory domain in the folded state[9]. The mechanism by which the folded HD inhibits catalytic activation has remained undefined, and one of two distinct possibilities exists: (i) the HD alters the positioning of bound $NAD^+$ to disfavor efficient catalysis or (ii) the HD blocks $NAD^+$ binding altogether.

Thus, despite prior insights into PARP-1 allostery and dynamics, there remain key questions regarding the regulation of PARP-1 activity, including the mechanism of HD-mediated autoinhibition, the details of PARP-1 interaction with $NAD^+$, and potential connections between the catalytic active site and the PARP-1 allosteric activation network. To address these questions of PARP-1 access and binding to substrate $NAD^+$ and the HD autoinhibitory mechanism, and to provide insights into PARP-1 regulation and dynamics, we have explored non-hydrolyzable $NAD^+$ analogs to directly assess the binding and accessibility of the PARP-1 active site during basal and DNA damage-activated conditions. Using x-ray crystallography, binding studies, catalytic activity assays, and HXMS, we find that the autoinhibitory HD of PARP-1 selectively restricts access to the $NAD^+$-binding site and provides a dynamic structure that can regulate the frequency of $NAD^+$ binding and utilization events. PARP-1 interaction with DNA strand breaks leads to full $NAD^+$ access to the catalytic active site, whereas PARP-1 alone has low frequency $NAD^+$ access that accounts for the basal level of unstimulated PARP-1 activity. Interestingly, we observe that $NAD^+$-binding site occupancy can influence the PARP-1 allosteric activation mechanism

such that affinity for DNA strand break damage is increased, providing new insights into PARP-1 domain assembly on DNA damage, and into how engagement of the PARP-1 allosteric activation network can influence persistence at sites of DNA strand breaks.

## Results

**PARP-1 CAT interaction with $NAD^+$ analogs**. We began our analysis with two $NAD^+$ analogs: carba-$NAD^+$ and benzamide adenine dinucleotide (BAD)[10, 11] (Fig. 1b). In carba-$NAD^+$, a carbon replaces the oxygen atom on the nicotinamide ribose, with the potential to alter the ribose conformation such that local geometry is no longer suitable for reaction chemistry. In BAD, carbon replaces the nitrogen atom in the nicotinamide ring, thus making a benzamide moiety (Fig. 1b), which is a poor leaving group. We analyzed whether these $NAD^+$ analogs could inhibit PARP-1 catalysis of PAR production, as would be expected for a non-hydrolyzable $NAD^+$ analog that is capable of binding. Indeed, BAD inhibited DNA-dependent PARP-1 automodification activity starting at a concentration equimolar with $NAD^+$ (50 μM) in a concentration series (Fig. 1c). In contrast, carba-$NAD^+$ exhibited only modest signs of inhibition at the highest concentration assessed (685 μM), suggesting that carba-$NAD^+$ was not efficiently binding to PARP-1.

To further address PARP-1 binding to the $NAD^+$ analogs, we used differential scanning fluorimetry (DSF) to measure the apparent melting temperature ($T_M$) of the PARP-1 CAT in the absence or presence of carba-$NAD^+$ and BAD, with the expectation that analog binding would lead to a relative increase in $T_M$. Notably, neither BAD nor carba-$NAD^+$ changed the apparent $T_M$ of PARP-1 CAT (Fig. 1d; Supplementary Table 1), suggesting that these compounds do not bind to the CAT. We next tested whether the $NAD^+$ analogs influenced the $T_M$ of PARP-1 CAT with the HD autoinhibitory domain deleted (ΔHD), which leads to constitutive activation and mimics HD unfolding during DNA-dependent PARP-1 activation[9]. Indeed, BAD substantially increased the $T_M$ of CAT ΔHD (10 °C increase at the highest concentration; Fig. 1d). In contrast, carba-$NAD^+$ had little influence on the $T_M$ for CAT ΔHD (<1 °C increase at the highest concentration), thus suggesting together with the inhibition results (Fig. 1c) that carba-$NAD^+$ does not bind to PARP-1 with appreciable affinity, and is poorly suited as a tool for understanding PARP-1 function.

PARP-1 CAT did not show evidence of binding to BAD in the DSF assay; however, benzamide alone increased the $T_M$ of both CAT WT and CAT ΔHD (Fig. 1e; Supplementary Table 1), indicating that benzamide binds to both proteins. The DSF results thus suggested that the HD presents a selective steric block to the PARP-1 active site, where a small compound such as benzamide is allowed access, but a larger compound such as BAD is not allowed to bind to the active site. In contrast to the benzamide moiety of BAD, the ADP-ribose moiety showed no evidence of binding to either CAT WT or CAT ΔHD, indicating that the benzamide portion provides the majority of binding affinity, and ADP-ribose has only a minor role in overall binding affinity. In confirmation, we compared the ability of ADP-ribose, benzamide, and BAD to inhibit the DNA damage-dependent activity of full-length PARP-1, and showed that ADP-ribose does not inhibit the reaction, whereas benzamide alone inhibits to a similar extent as the entire BAD molecule (Fig. 1f).

**Crystal structure of PARP-1 CATΔHD in complex with BAD**. We determined the x-ray crystal structure of the non-hydrolyzable $NAD^+$ analog BAD in complex with the constitutively active CAT ΔHD, thus providing the most

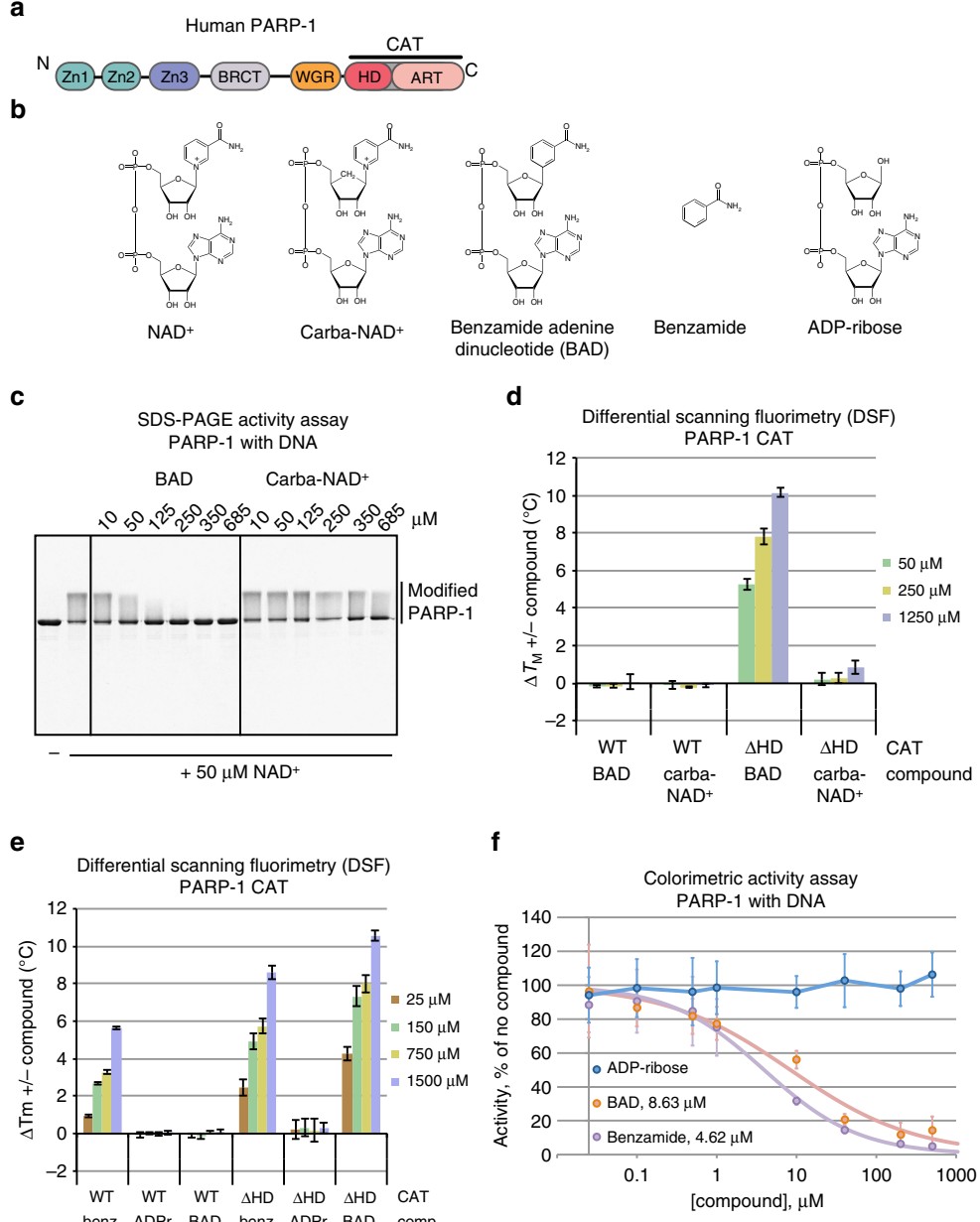

**Fig. 1** Non-hydrolyzable NAD[+] analog binding and inhibition of PARP-1. **a** Schematic representation of PARP-1 domains. **b** Chemical structure of key compounds used in this study: NAD[+], non-hydrolyzable NAD[+] analogs carba-NAD[+] and benzamide adenine dinucleotide (BAD), benzamide, and ADP-ribose (ADPr). **c** SDS-PAGE PARP-1 activity assay (1 μM DNA, 1 μM protein, 50 μM NAD[+]) in the presence of carba-NAD[+] and BAD. An image of the entire gel is included in Supplementary Fig. 10. **d, e** Differential scanning fluorimetry (DSF) experiment using PARP-1 CAT domain WT or ΔHD (5 μM) and various amounts of carba-NAD[+], BAD, benzamide, and ADP-ribose. $\Delta T_M$ is calculated by subtracting the $T_M$ and $\Delta T_M$ obtained in the absence of compound from the $T_M$ obtained in the presence of compound. The experiments were all performed in triplicate for which averages and standard deviations are shown. **f** Colorimetric assay of full-length PARP-1 DNA-dependent activity in the presence of ADP-ribose, benzamide, and BAD using 20 nM PARP-1, 40 nM DNA (18-bp duplex) and 50 μM NAD[+] (99:1, NAD[+]:bio-NAD[+]). Experiments were performed in quadruplicate and the averages and standard deviations for each compound concentration are shown, as well as the average IC50 values for benzamide and BAD

comprehensive view to date of PARP-1 active site interactions (Fig. 2a, b). The structure was determined by molecular replacement with diffraction data extending to 2.3 Å (Table 1). The asymmetric unit contains four CAT ΔHD molecules (A, B, C, and D). Each CAT ΔHD molecule has BAD bound in the active site (Fig. 2c). BAD adopts two conformations in molecules A, B, and D, with each conformation at a similar level of partial occupancy around 50% (Supplementary Fig. 1a). Molecule C exhibits a single conformation of BAD (Fig. 2a). The benzamide riboside moiety and the adenine riboside (adenosine) moiety adopt similar

conformations and form the same interactions with PARP-1 in all BAD molecules, whereas the positions of the two phosphate groups vary somewhat and account for the different BAD conformations (Fig. 2c). The adenosine phosphate and the benzamide riboside phosphate adopt a single conformation in molecule C (Fig. 2a, b). Molecules A, B, and D exhibit the same BAD conformation observed in molecule C, but each of these molecules also exhibit a second conformation in which the major difference is that the benzamide riboside phosphate rotates away from the benzamide ring (Fig. 2c; Supplementary Fig. 1a, b).

We expect that the single conformation observed in molecule C, and also in one of the conformers in each of the other molecules, likely represents the conformation assumed by NAD+ during the PARP-1 catalytic reaction based on comparison to ADP-ribosyl transferase toxins bound to NAD+ (Supplementary Fig. 1c, d).

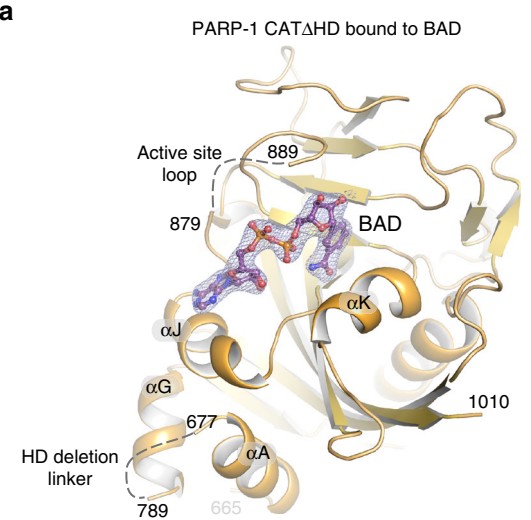

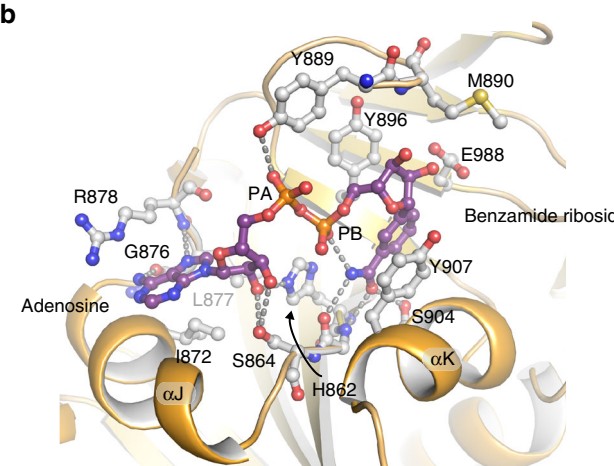

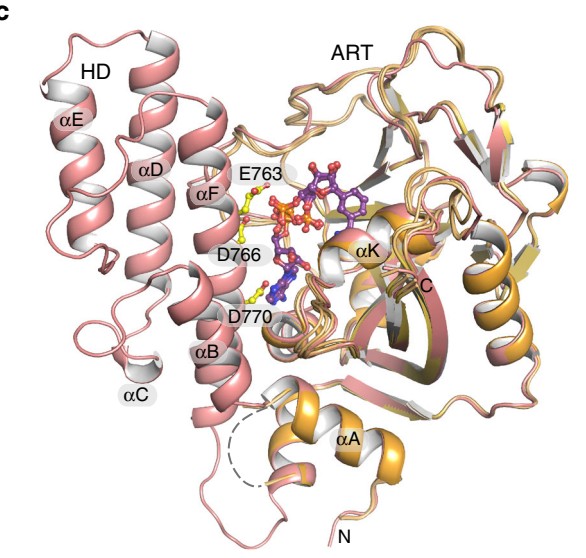

The active site loop (ASL; residues 879 to 889, also known as the D-loop) exhibits slightly different conformations in the crystal structure between the four different molecules (Supplementary Fig. 1b). The ASL is normally stabilized through contacts with the HD; however, the CAT ΔHD construct removes these contacts and appears to allow the ASL to assume a more flexible structure. Indeed, molecule C only exhibited weak electron density that did not permit modeling of the ASL, except for residue Y889 that forms a contact with BAD in each of the CAT ΔHD molecules (Fig. 2a, b). Based on our previous mutagenesis experiments and analysis of PARP-1 dynamics[9], the ASL is not a critical determinant of PARP-1 interaction with NAD+ but rather has a structural role in supporting contact between the ART fold and the HD.

Alignment of the CAT ΔHD/BAD structure with the structure of CAT WT indicates that the HD in its folded conformation would clash with bound BAD/NAD+. In particular, residues E763, D766, and D770 are positioned to interfere with the binding site of the adenine base and the phosphate groups of NAD+ (Fig. 2c), suggesting a potential mechanism to block substrate binding.

To date, PARP-1 interaction with NAD+ has been modeled based on the structures of related ADP-ribosylating toxins[12]; however, the CAT ΔHD/BAD structure provides a clear advance over modeling studies and provides an accurate model for the PARP-1 NAD+-binding site and how this site might be exploited in inhibitor design.

**DNA-dependent PARP-1 interaction with BAD.** CAT WT and CAT ΔHD served as models for inactivated and activated PARP-1. Therefore, we next tested whether full-length PARP-1 was capable of binding BAD, or whether interaction with DNA was required for BAD binding. As shown previously, PARP-1 exhibits a decrease in relative thermal stability upon binding to DNA as measured by DSF[8] (Supplementary Fig. 2, see WT), consistent with destabilizing HD structural changes and the HD unfolding mechanism identified using HXMS[9]. In line with the CAT ΔHD results, an increase in the $T_M$ of full-length PARP-1 owing to BAD was observed only with DNA present (Fig. 3a; Supplementary Table 1). A PARP-1 double mutant D766/D770A and triple mutant E763/D766/D770A were also tested in the DSF assay, as these residues clash with BAD in our molecular model (Fig. 2c), and both mutants have increased DNA-independent activity relative to PARP-1 WT[9] (Supplementary Fig. 3). Both mutants exhibited an increase in $T_M$ in the presence of BAD, and the observed increase did not require the presence of DNA (Fig. 3a), strongly suggesting a reduction in HD blockage such that the barrier for BAD binding to these mutants is lowered. We also tested mutants that inactivate PARP-1 through the disruption of interdomain contacts that form during PARP-1 allosteric activation[8]. These mutants are capable of binding to DNA, but they are unable to relay the DNA-binding signal to the HD

**Fig. 2** Crystal structure of PARP-1 CAT ΔHD bound to NAD+ analog BAD. **a** Crystal structure of the PARP-1 CAT domain carrying a HD deletion (residues 661–1011 with an 8-residue linker GSGSGSGG replacing residues 678–787) bound to BAD at 2.3 Å resolution. Four PARP-1 CAT ΔHD molecules were present in the crystal asymmetric unit; molecule C is shown. A 2.3 Å 2F_O−F_C weighted electron density contoured at 1.2 σ in the region of the bound BAD molecule is overlaid on the structure (see also Supplementary Fig. 1a, b). **b** Close-up view of the NAD+ binding site and contacts formed with the NAD+ analog BAD in molecule C. **c** The crystal structure of human PARP-1 CAT domain WT (pdb: 3gjw)[40] aligned with the PARP-1 CAT ΔHD/BAD structure

**Table 1 Crystallographic data and refinement statistics**

| | |
|---|---|
| Data collection[a] | |
| | PARP-1 CATΔHD with BAD |
| Space group | P1 |
| Unit cell dimensions | $a = 44.5$ Å $b = 75.4$ Å, $c = 85.6$ Å |
| | $\alpha = 98.7°$, $\beta = 104.9°$, $\gamma = 106.7°$ |
| | Four molecules/asymmetric unit |
| Wavelength (Å) | 1.01 |
| Resolution range (Å) | 50.0–2.3 (2.39–2.30) |
| Completeness (%) | 97.8 (97.3) |
| Average redundancy | 4.8 (4.9) |
| Mean $(I/\sigma I)$[b] | 13.2 (1.6) |
| $R_{merge}$ (%)[b] | 5.7 (111.7) |
| $R_{pim}$ (%)[b] | 2.9 (56.5) |
| Mean I CC(1/2)[b] | 0.999 (0.707) |
| Model refinement[a] | |
| Resolution range (Å) | 20.0–2.3 (2.36–2.3) |
| Number of reflections | 41,185 (3037) |
| $R_{cryst}$[c] | 0.199 (0.325) |
| $R_{free}$[c] | 0.222 (0.313) |
| Number of atoms/average | 7774/76.4 |
| B-factor (Å$^2$) | |
| Protein | 7499/76.8 |
| Solvent | 72/62.5 |
| BAD | 203/69.1 |
| Phi/Psi, preferred (%)/outliers (#)[d] | 97.6/0 |
| R.m.s.d. bond angles (°) | 1.488 |
| R.m.s.d. bond lengths (Å) | 0.010 |

[a] Values in parentheses refer to data in the highest resolution shell
[b] As calculated in SCALA[36]: $R_{merge} = \sum_{hkl}\sum_j |I_j - \langle I \rangle| / \sum_{hkl}\sum_j I_j$. $\langle I \rangle$ is the mean intensity of $j$ observations of reflection $hkl$ and its symmetry equivalents; $R_{pim}$ takes into account measurement redundancy when calculating $R_{merge}$; Mean I CC(1/2) is the correlation between mean intensities calculated for two randomly chosen half-sets of the data
[c] $R_{cryst} = \sum_{hkl} |F_{obs} - kF_{calc}| / \sum_{hkl} |F_{obs}|$. $R_{free} = R_{cryst}$ for 5% of reflections excluded from crystallographic refinement
[d] As reported in COOT[34]

domain due to inefficient formation of interdomain contacts. Domain interface mutations D45A (Zn1–WGR), W246A (Zn1–Zn3), and K633A (Zn3–WGR–HD) each exhibited a substantial reduction in $T_M$ shift in the presence of BAD and DNA (Fig. 3b; Supplementary Table 1). Correspondingly, each mutant also showed an absence or reduction in the $T_M$ decrease observed for PARP-1 WT owing to DNA-binding and allosteric signaling/ HD destabilization (Supplementary Fig. 2), thus connecting HD destabilization in the presence of DNA strand breaks to the ability to bind BAD.

PARP-1, PARP-2, and PARP-3 are the three human DNA-damage response PARPs whose catalytic activity is stimulated by DNA breaks, and each carries an HD. We tested whether PARP-2 and PARP-3 exhibited the same restriction on NAD$^+$ analog binding in the absence of DNA strand breaks. Indeed, PARP-2 and PARP-3 only showed evidence of BAD binding in the presence of activating DNA in the DSF assay (Fig. 3c; Supplementary Table 1). PARP-2 and PARP-3 required DNA bearing a 5' phosphate group (5'P), as these enzymes are selectively activated by this type of strand break[13, 14]. In contrast, the CAT of the PARP enzyme tankyrase 1 (TNKS/PARP-5a), which does not contain an HD, was capable of binding to BAD based on the DSF analysis (Fig. 3d; Supplementary Table 1). TNK1 required higher concentrations of BAD to observe a $T_M$ shift, consistent with the reported 1.5 mM $K_M$ of TNK1 for NAD$^+$[15]. Notably, there was no increase in $T_M$ for PARP-1 CAT containing the HD even in the presence of 4 mM BAD, suggesting that the HD does not simply lower the affinity for BAD/NAD$^+$ (Fig. 3d).

**ITC and NMR analysis of PARP-1 interaction with BAD.** Our DSF and crystallographic results suggested that PARP-1 is unable to bind BAD in the absence of DNA, and that an HD structural transition that occurs when PARP-1 engages a DNA break is necessary to remove residues blocking active site access. To confirm that the natively folded conformation of the HD imposes a substrate block, we used isothermal titration calorimetry (ITC) to directly measure PARP-1 binding to BAD. We first assessed CAT WT and CAT ΔHD interaction with BAD (Fig. 4a). Titration of BAD into CAT WT demonstrated no measureable heat exchange above the heat of dilution observed for BAD titrated into buffer (Fig. 4a). In contrast, PARP-1 CAT ΔHD demonstrated a clear heat exchange indicative of binding to BAD (Fig. 4a), with a $K_D$ of ~ 6 μM (Table 2).

As a control, we measured the ability of CAT WT and CAT ΔHD to bind to the smaller compound benzamide. Benzamide bound to both CAT WT and CAT ΔHD with a similar affinity in ITC experiments (Fig. 4b, c; Table 2), consistent with the DSF results and the model for a selective restriction on the PARP-1 active site. Moving to full-length PARP-1, there was no evidence of BAD binding in ITC experiments conducted without DNA (Fig. 4d). In contrast, PARP-1 in complex with a DNA strand break exhibited binding to BAD with an average $K_D$ of 1.8 μM (Fig. 4d; Table 2). Strikingly, full-length PARP-1 triple mutant E763/D766/D770A and double mutant D766/D770A showed evidence of binding to BAD in the absence of DNA (Fig. 4e and Supplementary Fig. 4; Table 2), consistent with DSF analysis and the crystal structure that indicated that these residues contribute to the HD steric block. Consistent with the model of selective active site access, PARP-1 WT bound to benzamide in the absence of DNA, and the E763/D766/D770A and D766/D770A mutants also bound to benzamide similarly (Supplementary Fig. 4). For E763/D766/D770A, the calculated $K_D$ for BAD is about 10-fold lower than observed for all other proteins. The increased affinity could potentially result from the E763/D766/ D770A mutation forming a pocket that accommodates the ADP-ribose portion of BAD and contributes in a positive manner to the binding interaction. We observed similar binding affinities across all of the ITC experiments; however, PARP-1 bound to DNA represents a complex system that could potentially have additional contributions to the measured heat, besides just ligand binding (i.e., changes in PARP-1 assembly on DNA upon BAD binding).

To provide further evidence that the PARP-1 active site is only available to NAD$^+$ analogs in the presence of DNA, we used one-dimensional $^1$H solution NMR spectroscopy to monitor BAD protons. A pulse sequence with a $T_{1\rho}$ filter was employed to reduce the signal contribution from non-exchangeable protons in the slower tumbling protein and DNA molecules (see Methods); however, the presence of folded protein was evident in data acquired with a pulse sequence lacking the $T_{1\rho}$ filter. NMR analysis of BAD alone at 20 μM yielded the expected spectrum with two notable peaks in the 8.05–8.35 ppm range (Fig. 4f), corresponding to the two non-exchangeable protons located on the adenine base (Supplementary Fig. 5). An essentially identical spectrum was measured when PARP-1 at 40 μM was included in the solution with BAD (Fig. 4f). When PARP-1 and DNA were included together with BAD, there was a significant decrease in the BAD spectrum peaks, consistent with BAD binding to PARP-1 and tumbling at a slower rate, thereby lowering the concentration of rapidly tumbling-free BAD. The level of signal reduction (~95%) is consistent with the expected amount of BAD to remain free in solution given the PARP-1/BAD concentrations used and the binding affinity measured by ITC. The NMR analysis further supports that PARP-1 does not interact with BAD until first interacting with DNA damage.

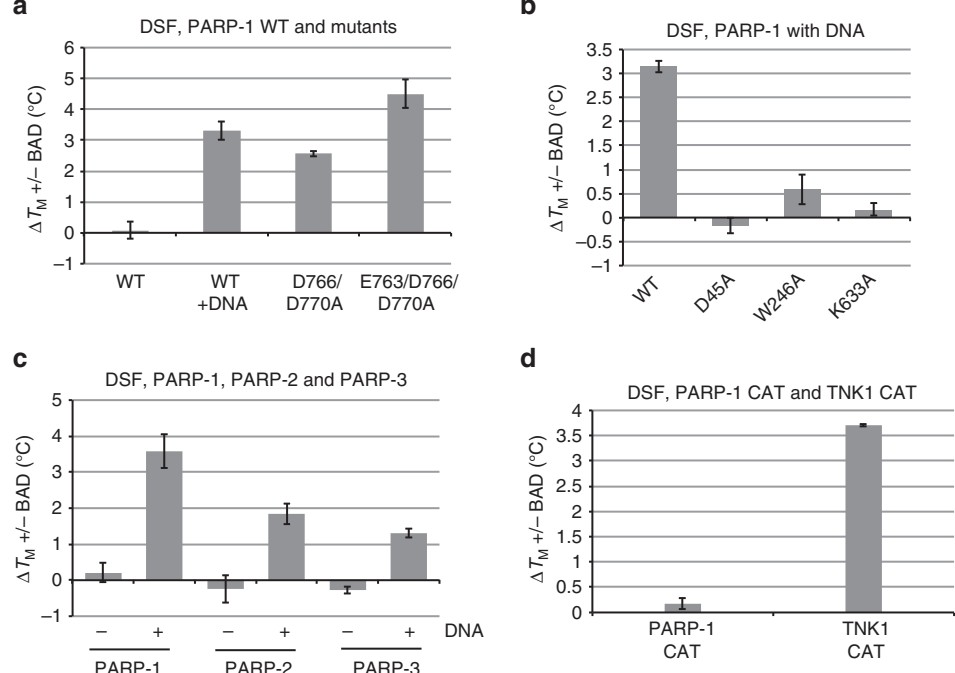

**Fig. 3** PARP-1 binds NAD$^+$ analog BAD only in the presence of DNA or when obstructing residues are removed. **a, b** DSF experiment using full-length PARP-1 WT or mutants (5 μM) in the presence or absence of DNA (2.5 μM, 26-bp duplex) and BAD (1250 μM). The $\Delta T_M$ is calculated by subtracting the $T_M$ obtained in the absence of compound from the $T_M$ obtained in the presence of compound. **c** DSF experiment using full-length PARP-1, PARP-2, and PARP-3 (5 μM) in the absence and presence of DNA (2.5 μM, PARP-1: 28-bp duplex, PARP-2: 28-bp duplex 5′P, PARP-3: 47-bp duplex with a central 5′P nick) and BAD (1250 μM). **d** DSF experiment using the CAT domains of PARP-1 and Tankyrase 1 (5 μM) with BAD (4 mM). The $\Delta T_M$ is calculated by subtracting the $T_M$ obtained in the absence of BAD from the $T_M$ obtained in the presence of BAD. The experiments were performed in triplicate and the averages and standard deviations are shown

**HXMS analysis of PARP-1 interaction with BAD**. We recently used HXMS to demonstrate DNA damage-dependent changes in PARP-1 dynamics[9]. HX measures amide proton exchange along the polypeptide backbone with deuterons in the buffer made with heavy water: HX is slow in stably folded regions (i.e., α-helices, interior of β-sheets), requiring transient local unfolding events to disrupt hydrogen bonds involving amide protons[16]. Thus, HXMS could provide complementary information to our crystallographic analysis of the PARP-1/BAD interaction (Fig. 2): HXMS is a solution-based method that can provide a view of local structure and dynamics, and that can access transient species that are too sparsely populated to be registered by bulk measurements produced by most conventional methodologies. Our prior published work using HXMS revealed major HX increases in HD helices αB and αF in response to DNA damage detection (i.e., HD unfolding, compare αB peptides in Supplementary Figs. 6 and 7)[9]. The HXMS experiments also indicated HX decreases in regions of PARP-1 involved in DNA binding and the formation of domain-domain interfaces, consistent with stabilization upon complex formation[9].

Here, we used HXMS to monitor BAD-dependent changes in dynamics for PARP-1 alone, in which the HD is in the folded state, or for PARP-1 in complex with a DNA single-strand break, in which the HD samples the unfolded state. We first recorded MS spectra at five time points of hydrogen exchange (Supplementary Fig. 8). The $10^2$ s time point revealed substantial changes in unfolded HD regions that are essentially at complete exchange at later time points, whereas the $10^4$ s time point revealed substantial changes in well-folded regions that showed minimal exchange at earlier time points. We thus focused our analysis on these two time points. The presence of BAD caused a decrease in HX for an ART active site peptide at the $10^4$ s time point,

consistent with BAD binding and stabilizing this region of the structure. This change in HX was only observed with PARP-1 in complex with DNA (Fig. 5a). In contrast, benzamide decreased HX in the same ART peptide for both PARP-1 alone and PARP-1 in complex with DNA, consistent with DSF and ITC results that indicated a selective restriction of BAD binding, and permissive binding of benzamide.

BAD-dependent HX changes in PARP-1 bound to a DNA strand break were not limited to the ART (Fig. 5b). The presence of BAD increased HX in peptides localized to HD helices αB and αF at the $10^2$ s time point (Fig. 5b, c). These HD regions undergo unfolding in response to DNA damage (αB and αF)[9], and they also interfere with the adenosine binding site of BAD/NAD$^+$ (αF; Fig. 2c). These HX changes are interpreted as BAD binding and further influencing the distribution of HD conformations between the unfolded state and folded state.

Interestingly, a BAD-dependent decrease in HX was also observed in regions outside of the CAT for PARP-1 in complex with DNA (Fig. 5b, c), but not in the absence of DNA (Supplementary Fig. 6). Strikingly, these changes localized to domain-domain (Zn1–Zn3, Zn1–WGR–CAT) and domain-DNA (WGR–DNA, Zn1–DNA) interfaces (Fig. 5c). These interfaces are formed in the presence of DNA and are critical for HD destabilization and PARP-1 activity[8, 9]. These HXMS results suggested a coupling between NAD$^+$ binding site occupancy and the multi-domain assembly of PARP-1 on DNA damage.

**NAD$^+$ binding influences PARP-1 interaction with DNA damage**. HXMS indicated that BAD binding further shifts the HD to the unfolded conformation, and that by supporting the unfolded HD conformation, BAD binding in effect stabilizes the

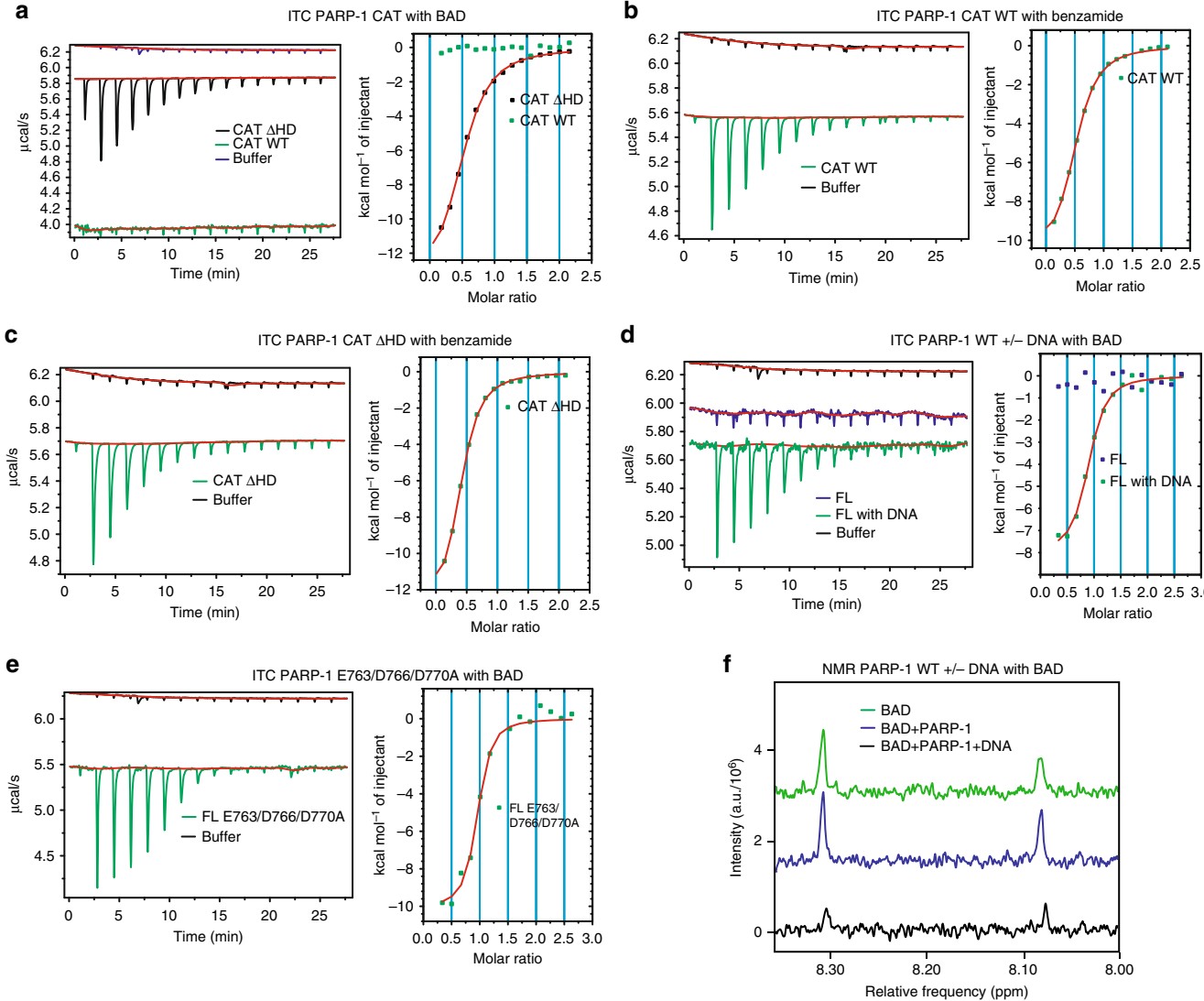

**Fig. 4** The autoinhibitory HD blocks NAD$^+$ binding to PARP-1. **a** ITC results where BAD (525 μM) was titrated into PARP-1 CAT WT and CAT ΔHD, both at 50 μM. **b, c** Same as in panel **a** but performed with benzamide as the titrant. **d** ITC results where BAD (525 μM) was titrated into full-length PARP-1 WT (40 μM) in the absence or presence of DNA (40 μM, 8 bp-duplex). **e** ITC results where BAD (525 μM) was titrated into full-length PARP-1 E763A/E766A/ D770A (40 μM). **f** NMR experiment showing the signal obtained for BAD (20 μM) alone in green, BAD (20 μM) with PARP-1 (40 μM) in blue, and BAD (20 μM) with PARP-1 (40 μM) and DNA (40 μM, dumbbell DNA containing a central 1 nt-gap) in black

---

**Table 2 Average binding affinities ($K_D$) determined by ITC**

| Protein | Compound | $K_D$ (μM) | n |
|---|---|---|---|
| CAT WT | BAD | No binding | – |
| CAT ΔHD | BAD | 5.87 +/− 0.62 | 0.63 +/− 0.09 |
| CAT WT | Benzamide | 3.79 +/− 0.98 | 0.49 +/− 0.02 |
| CAT ΔHD | Benzamide | 3.21 +/− 0.59 | 0.39 +/− 0.01 |
| FL WT | BAD | No binding | – |
| FL WT with DNA | BAD | 1.81 +/− 0.54 | 1.11 +/− 0.29 |
| FL D766/D770A | BAD | 2.49 +/− 0.02 | 0.96 +/− 0.10 |
| FL E763/D766/D770A | BAD | 0.38 +/− 0.23 | 1.00 +/− 0.13 |
| FL WT | Benzamide | 2.32 +/− 0.79 | 0.88 +/− 0.01 |
| FL D766/D770A | Benzamide | 2.81 +/− 0.11 | 0.81 +/− 0.14 |
| FL E763/D766/D770A | Benzamide | 2.99 +/− 1.76 | 0.89 +/− 0.07 |

The $K_D$ and stoichiometry (n) values were obtained by averaging the results of two or three independent ITC experiments. The reported errors are the calculated standard deviations (representing the range of values for experiments repeated two times)

interdomain contacts that are formed upon binding to DNA, and that drive HD destabilization in the first place. HXMS thus suggested that BAD binding might stabilize PARP-1 interaction with DNA by reinforcing the multi-domain assembly of PARP-1

on DNA strand breaks. We used a fluorescence polarization (FP) DNA competition assay to test the effect of BAD on PARP-1 interaction with DNA. PARP-1 was incubated with fluorescently labeled DNA in the presence or absence of BAD. Unlabeled DNA

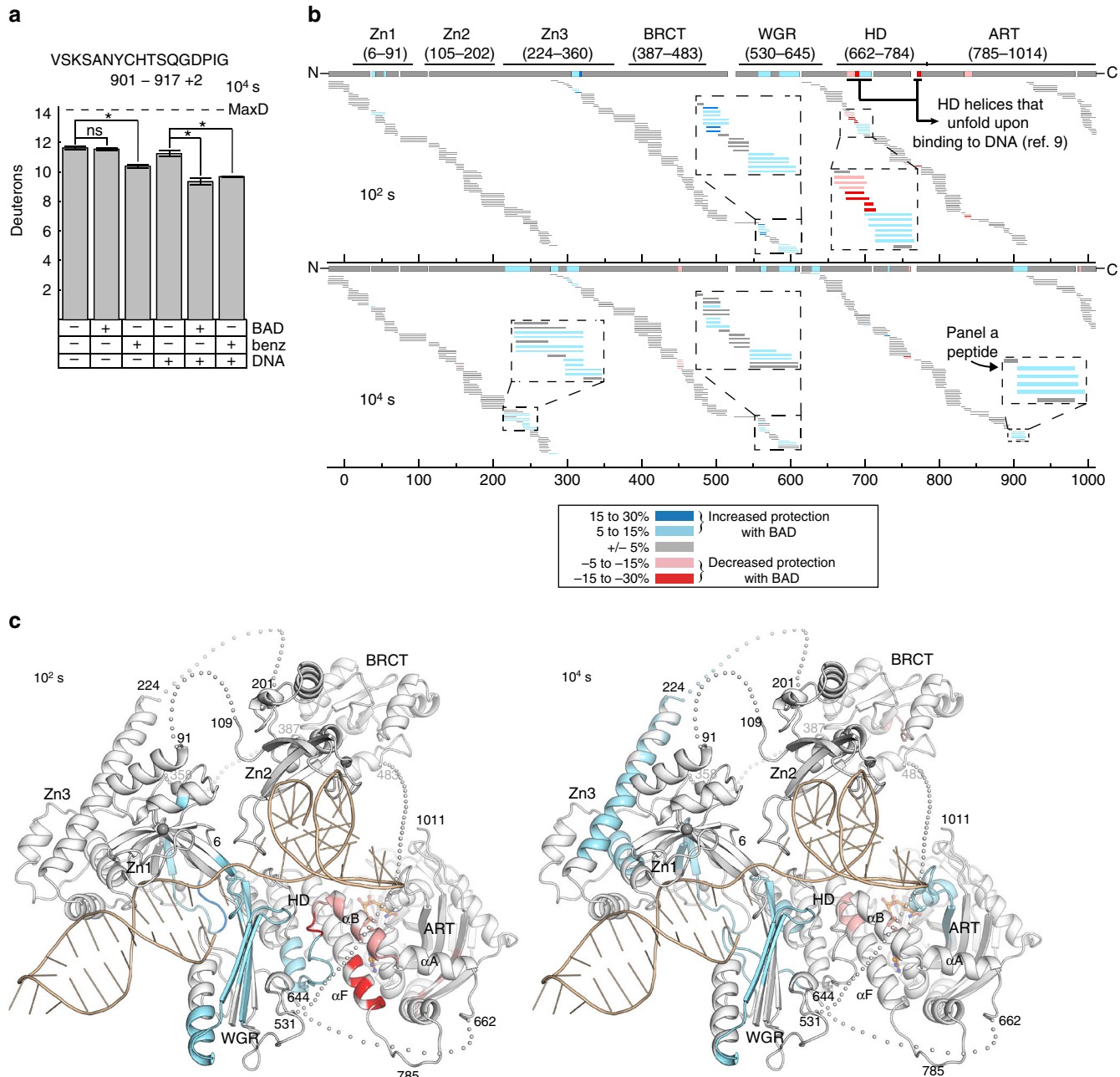

**Fig. 5** HXMS analysis of NAD$^+$ analog-dependent changes in PARP-1 dynamics. **a** Level of deuteration at $10^4$ s of a peptide from NAD$^+$ binding site (a.a. 901–917, charge state + 2) in the absence or presence of DNA (dumbbell DNA containing a central 1 nt-gap) and in the absence or presence of either BAD or benzamide. Errors were calculated as a standard deviation from three independent measurements. Maximum possible deuteration for the given peptide is shown by dashed line (MaxD). An asterisk denotes significant differences (*, $P < 0.0005$), with others marked as not significant (n.s.). **b** Percent difference in deuteration for each peptide, represented with horizontal bars, was calculated between DNA-bound PARP-1 samples in the absence and presence of BAD at $10^2$ s (top panel) and $10^4$ s (bottom panel) time points. Peptides that exhibit increased protection from deuteration upon BAD binding are shown in blue colors, whereas the ones with decreased protection from deuteration upon BAD binding are in red, as shown in the insert of the top panel. Each peptide color indicates the average percent difference in deuteration for the whole peptide, therefore there are some overlapping peptides where one exhibits difference in deuteration pattern, whereas the other one does not. The consensus behavior at each PARP-1 residue is displayed in the horizontal bar below the PARP-1 domain annotation. Gaps in protein coverage are indicated with white spaces. When available, data for multiple peptide charge states are presented. **c** The consensus for the HXMS data from panel **b** at $10^2$ s and $10^4$ s has been plotted on the combined crystallographic and NMR structure of PARP-1 in complex with a DNA single-strand break[6–8]. The linker residues (each shown as an individual sphere) and the BRCT (BRCA1 C-terminus) domain were manually positioned

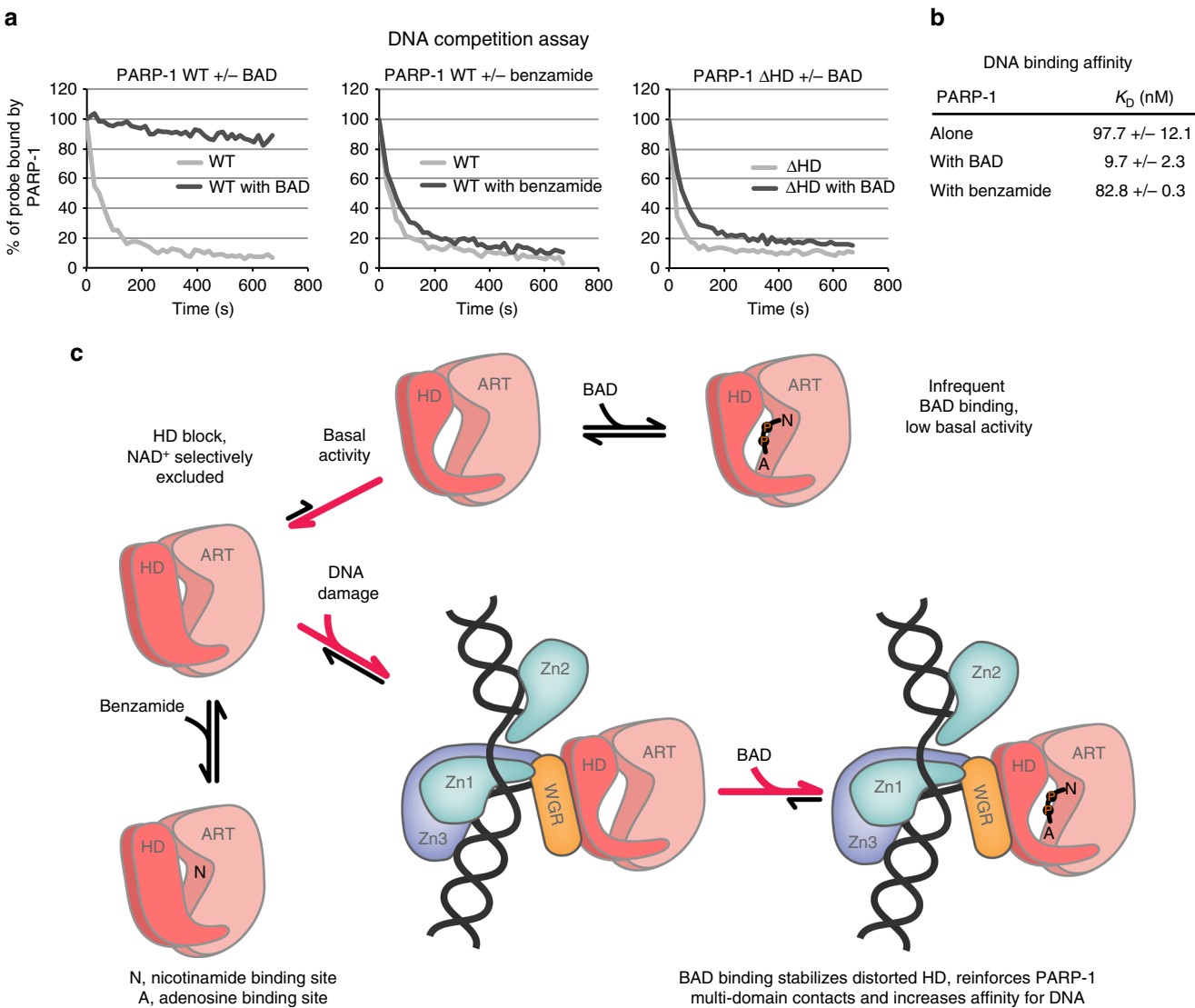

**Fig. 6** Allosteric coupling of PARP-1 catalytic active site and DNA-binding domains and model for the mechanism of HD regulation. **a** The fluorescence polarization DNA competition assay was performed by incubating 40 nM of PARP-1 WT or PARP-1 ΔHD with 20 nM of fluorescently labeled DNA (dumbbell DNA containing a central nick and an internal fluorescent FAM group) in the absence or presence of BAD or benzamide at 500 μM. A competitor unlabeled DNA was added and FP was measured over time. A representative experiment is shown. **b** PARP-1 affinity for DNA in the presence or absence of the indicated compounds, as measured by fluorescence polarization (see also Supplementary Fig. 9). **c** In its native state, the HD primarily exists in a folded conformation that selectively restricts access to the NAD$^+$-binding site, which can be subdivided into the nicotinamide binding site (N) and the adenosine binding site (A). Small molecules such as benzamide can access the NAD$^+$-binding site in the closed HD state, but BAD and NAD$^+$ binding is completely blocked in the folded state of the HD. The inherent dynamics of the HD sample the open conformation infrequently, leading to the basal level of PARP-1 catalytic output. The DNA-damage-dependent regulatory domains of PARP-1 form an interface with the HD that promotes an open, unfolded HD conformation that allows NAD$^+$ to have access to the active site. BAD binding to the catalytic active site shifts the HD equilibrium distribution to the unfolded state, thereby promoting the multi-domain contacts of PARP-1 with damage DNA, and increasing PARP-1 affinity for DNA (see panels **a** and **b**)

competitor was then added in excess and FP was measured over time, with an FP decrease reflecting PARP-1 exchange from labeled to unlabeled DNA. Without BAD, PARP-1 rapidly exchanged from labeled DNA to the unlabeled competitor DNA, suggesting that the multi-domain contacts of the PARP-1/DNA complex frequently "breathe" and provide opportunities for exchange from labeled to unlabeled DNA (Fig. 6a). Addition of BAD to the PARP-1/DNA complex greatly prevented the exchange of PARP-1 from the labeled to the unlabeled DNA (Fig. 6a). In contrast, benzamide did not affect PARP-1 release from the labeled DNA, indicating that catalytic active site occupancy alone cannot account for the stabilization observed with

BAD. The DNA release results are thus consistent with the HXMS data that showed a strengthening of interdomain and protein-DNA interactions in the presence of BAD, thereby creating resilience to competitor DNA.

Our model predicted that the strengthened interactions would depend on the ability of BAD to shift the distribution of HD conformations toward the unfolded state. To test this, we used a full-length PARP-1 mutant with a deletion of HD helices αD, αE, and αF (PARP-1 ΔHD) in the DNA competition assay. The deletion mutant is constitutively active in the absence of DNA damage[9]. Consistent with our model of BAD influence on the HD, BAD binding did not prevent the rapid release of full-length

PARP-1 $\Delta$HD from the labeled DNA, indicating that the HD is an essential aspect of the effect of BAD on PARP-1 allosteric signaling. The DNA competition experiment suggested a change in the equilibrium distribution of PARP-1 on and off DNA; therefore, we tested whether BAD influenced the apparent affinity of PARP-1 for DNA. Indeed, the presence of BAD led to a 10-fold increase in PARP-1 affinity for a DNA single-strand break, while the control compound benzamide had no effect (Fig. 6b and Supplementary Fig. 9). Together, our data demonstrate a direct allosteric connection between PARP-1 catalytic active site occupancy and PARP-1 engagement of damaged DNA (Fig. 6c).

## Discussion

Our study provides molecular level insights into a key aspect of PARP-1 regulation and function: $NAD^+$ utilization. We determined that BAD is an appropriate non-hydrolyzable $NAD^+$ analog for PARP-1 studies, and our analysis with PARP-2, PARP-3, and tankyrase suggests that BAD will be useful for the analysis of all PARP family members. In contrast, carba-$NAD^+$ does not efficiently bind to PARP-1, indicating that studies using this non-hydrolyzable analog for PARP-1 functional analysis should be interpreted with caution.

The CAT domain of chicken PARP-1 was crystallized in the presence of carba-$NAD^+$[17]. The structure showed no evidence of binding in the active site, which in light of our presented results is the expected outcome owing to the lack of robust PARP-1 binding to carba-$NAD^+$, and furthermore owing to the presence of the HD in a folded, substrate-blocking position. The chicken PARP-1 structure indicated a binding site for an ADP molecule, presumably representing an ordered portion of a bound carba-$NAD^+$ molecule. This binding site was defined as an acceptor site where ADP-ribose could potentially bind for subsequent addition of the next ADP-ribose molecule in a growing poly(ADP-ribose) chain. Our PARP-1/BAD structure indicates that this remains a plausible mechanism (Supplementary Fig. 1e); however, it will be necessary to form a ternary complex of PARP-1, BAD/$NAD^+$, and acceptor site ADP-ribose to understand the details of the chain elongation reaction. The modest amount of PARP-1 inhibition observed with carba-$NAD^+$ (Fig. 1c) could potentially result from a low level of binding to the acceptor site.

Comparing the CAT $\Delta$HD complex with BAD to structures of ADP-ribosyl transferase toxins in complex with $NAD^+$ demonstrates similarities in the compact conformation formed by the nicotinamide mononucleotide moiety that is proposed to support the transferase reaction[18–20] (Supplementary Fig. 1c, d). However, the comparison also illustrates that these toxin structures do not provide an accurate model for PARP-1 binding to $NAD^+$ (Supplementary Fig. 1c, d). For example, the structures of Exotoxin A[20] and diphtheria toxin[18] bound to $NAD^+$ provide perhaps the best models and are in overall good agreement with the structure of CAT $\Delta$HD bound to BAD; however, the adenine bases of the toxin structures are flipped 180° relative to the adenine base in the PARP-1/BAD structure (Supplementary Fig. 1c,d). Importantly, our crystal structure of the PARP-1/BAD complex represents the most accurate representation of $NAD^+$ binding to a PARP family member.

We initially envisioned two possible HD autoinhibitory mechanisms. In the first mechanism, the HD would intrude on the catalytic active site such that $NAD^+$ would bind in a sub-optimal conformation that could prevent efficient $NAD^+$ utilization, yet still allow low efficiency turnover to account for PARP-1 basal activity. The HD structural change would relieve this conformational strain and thereby lead to more efficient utilization of $NAD^+$ when PARP-1 engages DNA breaks. In the second mechanism, the HD in its folded state would entirely block $NAD^+$ binding to the active site. The low basal activity of PARP-1 would then result from the HD transiently sampling a conformation compatible with $NAD^+$ binding, and this conformation would be stabilized when PARP-1 engages DNA. Our findings provide strong evidence that the second scenario is the correct mechanism for HD autoinhibition of PARP-1.

Prior to our study, existing structural and biochemical studies were not able to distinguish between these two potential mechanisms. The CAT crystal structure of PARP-1 led to an early model for $NAD^+$-binding based on homology with diphtheria toxin and its complex with $NAD^+$[12]. This model did not indicate $NAD^+$ clashes with the PARP-1 HD (diphtheria toxin does not have an HD), suggesting that $NAD^+$ could indeed bind to the complete PARP-1 CAT. More recent studies of PARPs bound to inhibitors that extend toward the expected interaction site with the adenosine group of $NAD^+$[9, 21] indicated a potential clash between side chains emanating from the HD and the presumed binding configuration for the adenine base of $NAD^+$. However, it remained unclear whether these potential clashes would entirely prevent $NAD^+$ binding, or allow $NAD^+$ to bind but in a sub-optimal conformation; therefore it was not possible to distinguish the two proposed mechanisms. Our binding and HXMS studies with the $NAD^+$ analog BAD were required to resolve this issue, showing that the HD in its folded state entirely blocks $NAD^+$ binding.

Similar substrate-blocking mechanisms exist for other enzymes that use $NAD^+$ such as the $NAD^+$ glycohydrolase bacterial toxins from *Streptococcus pyogenes* (SPN) where an immunity factor binds SPN and completely blocks $NAD^+$ access to the active site as a way to regulate glycohydrolase activity[22, 23]. However, the anti-toxin appears to impose major changes on the $NAD^+$-binding site conformation, whereas the PARP-1 sub-strate-blocking mechanism of the HD does not influence the $NAD^+$-binding site conformation, but rather presents a steric block that can be more easily regulated, consistent with PARP-1 function in multiple cellular contexts and across different levels of activation.

Although the regulated hydrolysis of nucleotides that are pre-bound to enzymes is a commonly used mechanism of control in biology (e.g., GTPases, DNA helicases, DNA-sliding clamp loaders), our results indicate that PARP-1 is regulated through a complete block to $NAD^+$ binding. However, we picture the HD as a flexible gate to the $NAD^+$ binding site, where perturbations to the equilibrium distribution of HD conformations (i.e., folded vs. unfolded), either through native structural dynamics or under the influence of binding partners (e.g., PARP-1 regulatory domains, other cellular factors), allows $NAD^+$ access to the active site and thereby regulates PARP-1 catalytic output (Fig. 6c). In the absence of DNA breaks, the HD will only sample the unfolded conformation infrequently, thus supporting a low level of $NAD^+$ binding and basal activity. Upon binding to DNA breaks, the HD will transition to the unfolded conformation through interactions with the regulatory domains assembled on DNA (Fig. 6c). Molecules such as benzamide that only occupy the nicotinamide binding site (N) of the CAT and do not extend into the adenosine-binding site (A) are able to bind PARP-1 in the absence of DNA, as indicated by our data. As the folded HD provides a block to $NAD^+$ binding, we anticipate that all factors that regulate PARP-1 activity, such as phosphorylation, must act to promote the unfolded state of the HD, thereby increasing the frequency of $NAD^+$ binding to the active site.

What are the potential advantages of not having PARP-1 pre-loaded with $NAD^+$ in the cell? As PARP-1 is an abundant nuclear enzyme, a basal state in which PARP-1 is not bound to $NAD^+$ could mitigate its potential to influence the pool of free cellular $NAD^+$ that is not pre-bound as a co-factor to metabolic enzymes.

PARP-1 appears to have a fairly promiscuous ability to PAR modify target proteins with a variety of sequence contexts, with no clear consensus sequence for modification sites. Thus, PARP-1 pre-bound to $NAD^+$ would likely be primed for adverse off-target modification events. The HD autoinhibitory domain could therefore regulate $NAD^+$ access to PARP-1 in only the appropriate signaling environments related to DNA damage repair or transcriptional regulation.

Our observation that the occupancy of the $NAD^+$-binding site influences HD structure, interfaces between regulatory domains, and affinity for DNA strand breaks provides important insights into PARP-1 allosteric signaling and has interesting implications for PARP-1 function and inhibition. The results indicate that the distorted, activated conformation of the HD essentially contributes to the multi-domain assembly of PARP-1 on DNA damage. The HD thus can perform two functions depending on its conformation: (i) serve as a block to $NAD^+$ binding in the fully folded conformation or (ii) contribute to PARP-1 multi-domain binding to DNA in the distorted conformation. BAD binding positions the ADP-ribose moiety such that it pushes the HD toward the unfolded conformation that contributes to DNA binding, thus explaining the effect of BAD on PARP-1 interaction with DNA damage (Fig. 6a, b). Interestingly, the CAT of human PARP-2 was required for efficient recruitment to sites of DNA damage and high affinity interaction with DNA damage, and a mutation at the WGR-HD interface had a similar affect as CAT deletion[24], suggesting that the HD of PARP-2 might also contribute to interaction with DNA damage[24, 25].

Pommier and colleagues discovered that PARP inhibitors shift the nuclear distribution of PARP-1 from the soluble fraction to an insoluble, chromatin bound fraction, and they termed this effect "PARP trapping"[26]. Clinical PARP inhibitors trapped PARP-1 to varying degrees, leading Pommier and colleagues to hypothesize that the different binding modes of PARP inhibitors might influence PARP-1 allostery to varying degrees, and thus lead to a "reverse allostery" that could potentially influence PARP trapping on chromatin. An activity assay was used in their study to follow PARP-1 release from DNA upon modifying itself with PAR, and the tested clinical inhibitors showed different propensities to affect PARP-1 release after modification[26]. However, there have been no direct measurements of PARP allostery that connect catalytic active site binding to DNA-binding activity, and current structures of PARP-1 CAT bound to inhibitors do not provide an evident molecular explanation for the enhanced ability of certain inhibitors to trap PARP-1 on DNA. Thus, our study provides a direct testing of the "reverse allostery" hypothesis.

It is possible that $NAD^+$ binding to the catalytic active site could impact PARP-1 persistence at sites of DNA damage. PARP-1 production of PAR is required for the recruitment of repair factors, yet is also required for PARP-1 release from damage sites, indicating a balance between $NAD^+$ binding/PAR production and residence time at sites of DNA damage. Our finding that the occupancy of the $NAD^+$-binding site with BAD locks PARP-1 on a DNA break (Fig. 6) has clear implications for clinical strategies to inhibit PARP-1 since all inhibitors in the clinic target the $NAD^+$-binding site[27]. Thus, the allosteric changes and the effects on DNA binding conferred to PARP-1 could impact in diverse ways the behavior in the cell—and efficacy in the clinic—of each inhibitor.

## Methods

**Gene cloning and mutagenesis**. Full-length PARP-1 WT (residues 1–1014) was cloned in pET28 vector and site-directed mutagenesis was performed using QuikChange protocol (Stratagene)[28]. The PARP-1 CATΔHD construct used for crystallization was created by replacing residues 678–787 with an eight-residue linker (GSGSGSGG) in a pET28 construct coding for PARP-1 residues 661-1011[9].

The linker connects α-A to α-G, thus replacing the HD. The PARP-1 CAT WT (residues 661–1014) was cloned in pET28 vector[28]. PARP-2 (isoform 2, residues 1–570) was cloned in pET28 vector[13]. PARP-3 (isoform b, residues 1–533) was cloned in pDEST17 vector (gift from Dr. Ivan Ahel). Tankyrase 1 CAT (residues 1093–1327) was cloned in pET24.

**Protein expression and purification**. Full-length PARP-1 WT and mutants expression and purification were described previously[9, 28–30]. PARP-1 CAT WT and CATΔHD were expressed and purified as described[8, 9]. Note that for CATΔHD and PARP-1 overactive mutants D766/D770A and E763/D766/D770A, 10 mM benzamide was added to the E.coli medium to reduce cellular toxicity of the PARP-1 protein. Tankyrase 1 CAT was purified with a two-step protocol, using a $Ni^{2+}$ affinity column followed by a gel filtration column as described[28]. Purification of full-length PARP-1 ΔHD was performed as described[9].

**BAD and Carba-$NAD^+$ preparation**. BAD and carba-$NAD^+$ were synthesized by Viva Biotech Lmt. (Shanghai). Proton NMR spectra and liquid chromatography mass spectrometry confirmed the structures of the compounds. Prior to usage, the pH of the compounds was adjusted to pH 8.0 by adding Tris pH 8.8 to a final concentration of 30–40 mM.

**SDS-PAGE PARP-1 activity assay**. The SDS-PAGE activity assay was performed as described[28] using 1 μM protein, 1 μM DNA (18 bp oligonucleotide), 50 μM $NAD^+$, and various concentrations of BAD and carba-$NAD^+$ as indicated. Reactions were incubated for 5 min before quenching with SDS-PAGE loading buffer.

**DSF**. DSF experiments were performed as described[8, 13] using 5 μM protein, 2.5 μM DNA when indicated and various concentrations of compound (BAD, carba-$NAD^+$, benzamide, and ADP-ribose) as indicated in Fig. 1 and Fig. 3. Experiments were performed on a Roche LightCycler 480 RT-PCR. Reactions including DNA were performed with a 26- or 28-base pair (bp) unphosphorylated oligonucleotide for PARP-1, a 28-bp 5′-phosphorylated (5′P) oligonucleotide for PARP-2, and a 47-bp oligonucleotide including a central 5′P nick for PARP-3[13] in the following buffer: 25 mM Hepes pH 8.0, 150 mM NaCl, 1 mM EDTA, and 0.1 mM TCEP. Reactions with PARP-3 were conducted at lower ionic strength to allow binding to DNA as described (25 mM Hepes pH 8.0, 50 mM NaCl, 1 mM EDTA and 0.1 mM TCEP)[13]. $\Delta T_M$ values were calculated by subtracting the $T_M$ determined for the protein in the absence of compound from the $T_M$ determined in the presence of compound. Experiments were performed in triplicate and a Boltzmann sigmoid was fit to the data to determine the $T_M$ values (KaleidaGraph).

**Colorimetric PARP-1 assay**. The colorimetric assay was performed as described[8, 28] using 20 nM of protein and 40 nM DNA (18-bp duplex). A mixture of $NAD^+$ and biotinylated $NAD^+$ (bio-$NAD^+$) was used at a 99:1 ratio and a final concentration of 50 μM. Benzamide, ADP-ribose or BAD was added to the $NAD^+$ mix at the concentration indicated in Fig. 1d. Reactions were allowed to proceed for 5 min before quenching. Three independent experiments were performed.

**Crystallization and structure determination**. PARP-1 CATΔHD (30 mg per ml) was crystallized in the presence of 1.6 mM BAD in 24 to 29% PEG 3350, 0.2 M NaCl, 0.1 M Bis-Tris pH 5.5 in sitting drop vapor diffusion trays at room temperature (RT). Crystals were transferred to a stabilizing solution (23.3% PEG 3350, 0.1 M NaCl, 50 mM Bis-Tris pH 5.5, 1.6 mM BAD) and then to a cryo-protection solution (26% PEG 3350, 0.1 M NaCl, 50 mM Bis-Tris pH 5.5, 1.6 mM BAD, 10% sucrose) prior to flash-cooling in liquid nitrogen. X-ray diffraction data were collected at SIBYLS beamline 12.3.1 (Advance Light Source) and processed using XDS[31] (Table 1). The structure was determined by molecular replacement using PHASER[32] as implemented in the Phenix suite[33] and PDB code 5ds3[9] as a search model. Model building was performed using COOT[34] and refinement was performed using REFMAC5[35, 36]. Structure images were made using PYMOL Molecular Graphics System (Schrödinger, LLC).

**Isothermal titration calorimetry**. ITC experiments were performed using a MicroCal iTC200 calorimeter (GE Healthcare). BAD or benzamide at 525 μM was titrated into PARP-1 protein (50 μM for CAT WT and CAT ΔHD, 40 μM for full-length PARP-1 WT and mutants) or PARP-1 protein with DNA (20–45 μM) at 25° C. Prior to the experiment, proteins and DNA were dialyzed in 25 mM Hepes pH 8.0, 150 mM NaCl, 1 mM EDTA, and 0.1 mM TCEP. BAD and benzamide were diluted in the dialysis buffer. PARP-1/DNA complexes were assembled and incubated at RT for 30 min prior to performing the experiment. Experiments were typically designed as follows: a first injection of 0.2 μL was followed by 16 injections of 2.5 μL with a spacing of 100 s between injections. Excess heat upon each injection was integrated using an automatically adjusted base line and derived values corrected by heat of dilution. Data analysis and curve fitting were conducted using Origin 7 software. The ITC experiment for each sample was performed either two (PARP-1 WT with benzamide, PARP-1 E763/D766/D770A with benzamide, PARP-1 D766/D770A with BAD) or three times (the remaining eight samples). Table 2 lists the average $K_D$ value and stoichiometry ($n$) for the replicates, and the

reported errors are the calculated standard deviations (representing the range of values for experiments repeated two times). We anticipate that the sub-stoichiometric $n$ values for CAT constructs results from either the active fraction being < 100%, or the protein concentration determination being not entirely accurate. The analysis of full-length PARP-1 yielded $n$ values around the expected value of 1.

**NMR**. Full-length PARP-1 WT and a dumbbell DNA containing a central 1-nucleotide gap (5′-GCTGGCTTCGTAAGAAGCCAGCTCGCGGTCAGCTT GCTGACCGCG-3′)[7], were dialyzed against a modified PBS buffer (222 mM NaCl, 2.7 mM KCl, 200 μM DTT, 10 mM sodium phosphate, pH 7.4). An aliquot of the dialysis PBS buffer was then lyophylized and resuspended in deuterated water (D$_2$O). The NMR samples were prepared by diluting the protein (40 μM final), DNA (40 μM final), and BAD (20 μM final) in the deuterated buffer such that all samples contained 85% D$_2$O final. The samples were incubated 30 min at RT prior to collecting the NMR spectra. $^1$H (700.303 MHz) NMR spectra were acquired with a Bruker Avance II 700 spectrometer equipped with a 16.4 T narrowbore magnet and a double-resonance ($^1$H/$^{13}$C) cryoprobe. All samples were run using 3 mm NMR tubes (180 μL). 1D experiments were acquired with excitation sculpting to suppress residual water signal[37]. The use of D$_2$O as the primary solvent (85%) removed the signals from amide hydrogens in the region of interest (7–9 ppm), however a low-power $T_{1\rho}$ filter (70 ms) was necessary after the 90-degree excitation pulse to reduce signals from the non-exchangeable hydrogens arising from the large, slowly tumbling protein and DNA.

**HXMS measurement and analysis**. Deuterium on-exchange was carried out at RT by mixing 5 μL of each sample (8 μM concentration) with 15 μL of deuterium on-exchange buffer (10 mM HEPES, pD 7.0, 150 mM NaCl, in D$_2$O) yielding a final D$_2$O concentration of 75%. pD values for deuterium-based buffers were calculated as pD = pH + 0.41[38]. Upon mixing with on-exchange buffer, the amide protons will be replaced over time with deuterons yielding an increased peptide mass. To quench the deuterium exchange reaction, the sample (20 μl) was mixed with 30 μl of ice-cold quench buffer (1.66 M guanidine hydrochloride, 10% glycerol, and 0.8% formic acid, for a final pH of 2.4–2.5) and rapidly frozen in liquid nitrogen. For the MS analysis, each sample (50 μL) was melted on ice and loaded onto an in-house packed pepsin column for digestion. Pepsin (Sigma) was coupled to POROS 20 AL support (Applied Biosystems), and the immobilized pepsin was packed into a column housing (2 mm × 2 cm, Upchurch). Pepsin-digested peptides were captured on TARGA C8 5 μm Piccolo HPLC column (1.0 × 5.0 mm, Higgins Analytical) and eluted through an analytical C18 HPLC column (0.3 × 75 mm, Agilent) by a shaped 12–100% buffer B gradient at 6 μL per min (Buffer A: 0.1% formic acid; Buffer B: 0.1% formic acid, 99.9% acetonitrile). The effluent was electrosprayed into the mass spectrometer. Non-deuterated (ND) PARP-1 samples were prepared in 10 mM HEPES, pH 7.0, 150 mM NaCl buffer, and mixed with the quench buffer to mimic the samples from on-exchange reaction. Peptides of ND samples were analyzed in tandem MS (LTQ Orbitrap XL, Thermo Fisher Scientific). We analyzed MS/MS data collected from these samples to identify potential PARP-1 peptides using SEQUEST (Bioworks v3.3.1, Thermo Fisher Scientific) with a peptide tolerance of 8 ppm and a fragment tolerance of 0.1 AMU. Deuterated samples were then analyzed on an Exactive Plus EMR-Orbitrap (Thermo Fisher Scientific). A MATLAB based program, ExMS[39] was used for HXMS data processing. ExMS uses the list of MS/MS peptides found by the SEQUEST search, identifies individual isotopic peaks and peptide envelopes, and calculates centroid values of these peptide envelopes. ExMS first analyzes the non-deuterated sample to identify the peptide envelope centroid values as well as the chromatographic elution time ranges of each parental non-deuterated peptide. ExMS then uses the information from the non-deuterated analyses to identify deuterated peptides in each sample of the HXMS timecourse. The level of deuteration of individual peptides was measured by comparing them to the ND sample. Each individual deuterated peptide is corrected for loss of deuterium label during HXMS data collection (i.e., back exchange after quench) by normalizing to the maximal deuteration level of that peptide, which we measure in a "fully deuterated" (FD) reference sample. The FD sample was prepared in 75% deuterium to mimic the exchange experiment, but under acidic denaturing conditions (0.5% formic acid), and incubated over 48 h to allow each amide proton position along the entire polypeptide to undergo full exchange. For each peptide, we compare the extent of deuteration as measured in the FD sample to the theoretical maximal deuteration (i.e., if no back-exchange occurs); the median extent of back-exchange in our datasets is only 15%.

**HXMS plotting**. Peptide plotting and calculations were performed in MATLAB. Plots reporting percent difference in HX (i.e. Fig. 5b) include all peptides for which the identical peptide was found in both conditions (e.g., PARP-1/DNA complex with and without BAD), the ND, and FD samples. For comparing two different HXMS datasets, we plot the percent difference of each peptide, which is calculated by subtracting the percent deuteration of one sample from that of another, and plotted according to the color legend in stepwise increments (as in Fig. 5b). Consensus behavior at each residue was calculated as the average of the differences in HX protection of all peptides spanning that residue. We include in our figures

peptides of identical sequence but different charge states. Although not unique peptides, they do add confidence to our peptide identification as their deuteration levels are in close agreement with each other. The HXMS peptides plotted to span the length of PARP-1 all come from one experiment collected within a 24 h window in order to avoid slight changes in fragmentation patterns between independently performed experiments that were collected at different points in time. This choice in presentation allows us to maximize the number of peptides (identified by both a given sequence and charge state) that we can identify in both the non-deuterated and fully deuterated samples as well as between conditions across time points. Nonetheless, the findings reported in this paper regarding the dynamics of each of the domains of PARP-1 are representative of the results of multiple experiments. For the plot of peptide data expressed as the number of deuterons, the values are expressed as the mean of three independent measurements + / − s.d. The actual HXMS of each peptide, rather than the difference in deuteration between different conditions, is shown in Supplementary Figs 6 and 7.

**Fluorescence polarization**. For the DNA competition assay, 40 nM PARP-1 WT, or PARP-1 ΔHD, was incubated with 20 nM of dumbbell DNA with a central nick carrying an internal fluorescent FAM group (5′-GCT GAG C/FAMT/T CTG GTG AAG CTC AGC TCG CGG CAG CTG GTG CTG CCG CGA-3′) for 30 min at RT in 12 mM Hepes pH 8.0, 60 mM KCl, 8 mM MgCl$_2$, 4% glycerol, 5.7 mM beta-mercaptoethanol, 0.05 mg per ml bovine serm albumin with or without BAD or benzamide (500 μM). A competitor unlabeled DNA of the same sequence was added at 100 nM and FP was measured over time on a VictorV plate reader (Perkin Elmer). For the DNA affinity measurement assay, increasing concentrations of PARP-1 WT were incubated for 30 min at RT with 5 nM of dumbbell DNA probe carrying a nick (described above) in the absence or presence of BAD or benzamide (100 μM) in the following buffer: 12 mM Hepes pH 8.0, 250 mM NaCl, 4% glycerol, 5.7 mM beta-mercaptoethanol, 0.05 mg per ml bovine serum albumin.

**Data availability**. The atomic coordinates and structure factors have been deposited in the Protein Data Bank under accession code 6BHV. Other data are available from the corresponding authors upon reasonable request.

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

## Acknowledgements

We acknowledge support from the NIH (R01GM087282 to J.M.P. and B.E.B. and R01GM105654 to B.E.B.) and the CIHR (BMA342854 and PJT374609 to J.M.P.). Efforts to apply crystallography to characterize eukaryotic pathways relevant to human cancers are supported in part by National Cancer Institute grant Structural Biology of DNA Repair (SBDR) CA92584. A portion of the work was conducted at the Advanced Light Source (ALS), a national user facility operated by Lawrence Berkeley National Laboratory on behalf of the Department of Energy, Office of Basic Energy Sciences, through the Integrated Diffraction Analysis Technologies (IDAT) program, supported by DOE Office of Biological and Environmental Research. Additional support comes from the National Institute of Health project MINOS (R01GM105404) and a High-End Instrumentation Grant S10OD018483.

## Author contributions

M.F.L. performed protein purification, binding studies, activity assays, and X-ray crystallography. L.Z. performed HXMS experiments and analysis. P.M.A. performed NMR experiments and analysis. M.F.L. and J.M.P. wrote the manuscript with input and contributions from all authors. J.M.P. and B.E.B. directed the research.

## Additional information

**Competing interests:** The authors declare no competing financial interests.

