## [Peer Review File · Nature Communications]

Reviewers' comments:

Reviewer #1 (Remarks to the Author):

In the manuscript “Non-hydrolyzable NAD⁺ analog reveals substrate blocking mechanism for PARP-1 and reverse allosteric communication from catalytic center to DNA binding domains” Langelier and colleagues explore the structural basis of PARP-1 activation mechanism. The manuscript describes precisely for the first time how NAD is bound to the active site of PARP-1. Furthermore, it clarifies the intramolecular regulation of PARP-1 via an allosteric block by catalytic auxiliary domain HD and PARP-1 stabilisation at the DNA breaks upon NAD binding. These data will also have important implications for understanding how the clinically relevant PARP inhibitors work. The manuscript is nicely written and provides some very important new insights into the complex mechanism of PARP-1 activation. I suggest several small changes/additions to further improve the manuscript:

DSF data:

It would be nice if the authors could provide T_m's of their reference proteins (e.g. in a supplementary table) so the reader can evaluate the dT_m values based on these reference temperatures. Also it would allow to assess the influence of the point mutation on the thermal stability of the protein.

BAD dual conformers:

- i) It would be helpful to the reader if the authors would present electron density maps for the BAD dual-conformers (or at least one of them) and give estimated ratios of these conformers for each of the subunits (A-D).
- ii) On page 7 the authors state that “the second conformation of BAD in molecule B appears to be influenced by crystal packing” and concluded that the single conformation of subunit C is the active conformation. No evidence is presented for this assumption and it would be great if the authors could elaborate on this point as the exact binding of NAD is crucial to the understanding of PARP-1's activation mechanism.

ITC data:

- i) The authors demonstrate by HXMS and fluorescence polarisation that binding of BAD to full-length PARP-1 influences DNA binding and via structural re-arrangements in Zn3, BRCT, WGR and HD/ART. The authors should consider that not all of these structural changes occur at the BAD binding site. This structural changes will, therefore, contribute to the measured heat in the ITC experiment, thus contributing to the calculated K_d value, without being a direct consequence of contacts established between ligand and protein. The authors should comment this.
- ii) Figure 4e: One data point is at a molar ration <0. It would be good if the authors could double check their data.

Page 5 last line – Figure reference should be to Fig. 1d.

It would be nice if the authors could discuss whether similar mechanisms to one described here can be found in other proteins.

Reviewer #2 (Remarks to the Author):

Reviewer comments

The manuscript by Langelier et al. reports a crystal structure of PARP-1 catalytic domain with a substrate analog and describe a two-way communication between the catalytic C-terminal ADP-ribosylating domain and the N-terminal DNA binding domains. Despite significant efforts by multiple research teams there are no substrate bound structures for the enzymes of the PARP family and the structure reported here is the closest one to the actual NAD⁺ substrate. The study itself is adding on to previous work published by the authors where they have studied activation mechanism of the PARP1 enzyme by binding to DNA. The manuscript provides further evidence for the allostery, but will also explain earlier observations through detailed analysis of the protein. The manuscript is fluent and well written and will increase our understanding on how the key enzyme in DNA damage recognition works.

My comments to the author are listed below:

Page 5 and Figure 1c

The authors have used chicken PARP1 in structural comparison and also describe binding of partial carba-NAD⁺ to this particular structure. This carba-NAD⁺ fragment has been thought to mark the acceptor site as mentioned and it should be added to the figure. It is not entirely clear why chicken structure was selected as there are crystal structures of the human enzyme also available. There are residues differences (E/Q763) and this should be explained in the figure legend.

Authors should explain, why in Fig. 1a (monomer C) the active site loop is not visible, while it is present in the other monomers having the same BAD binding mode (Figure S1). Despite this the binding mode in monomer C would be the catalytically relevant mode.

Are there any interactions of BAD and the D-loop/ASL and or is it just disordered upon binding. In the crystal structure used as a model in molecular replacement the ASL is visible. Are there any changes observed between the structures? This could be added as a supplementary figure.

Now the authors have solved a co-crystal structure of close substrate analog yet they do not discuss the catalytic mechanism at all, which is unsatisfying in the structural analysis. Is the carba-NAD⁺ fragment at all in a suitable position for chain elongation?

“engineered link” is included in the Figure 1 a, but it is not defined in the legend or text and the rationale for the engineering is not given in the methods. This should be clarified.

Please define in figure legend the “DNA”. This is different for the enzymes and would help the reader to understand the experiment when looking at the figures.

page 6

IC50 curves in Figure 1f should be fitted. How come the IC50 is so much higher than the Kd measured with ITC?

page 7

Before stating that there is a single conformation in molecule C it should be stated that there are multiple conformations for the BAD. Now it is difficult to grasp when reading the first time what is meant.

page 9

Authors refer to an old publication, by Rippmann et al. which measured a Km of TNKS1 to be 1.5 mM. In a recent article Thorsell et al. (PMID: 28001384) measured a Km for TNKS1 to be very similar to that of PARP1. Therefore this rationale may not be valid. One can not effectively measure affinities of a compound to different proteins using DSF. Could it be that the deltaTm for TNKS1 is just overall smaller when substrate binds? It appears as if a concentration series was measured and then it would be possible to fit this and get an estimated Kd (Alternatively ITC could be used) Authors should also list the actual Tm values for the proteins at least in the supplementary file and not just the delta.

Text mentions 4 mM BAD and figure legend 1.25 mM.

Define the errors in Table 2.

page 16

Comparison of the NAD+ binding to toxins and to PARP1 should be shown.

page 19

There is also a study showing potential contribution of the CAT to the DNA binding (Obaji et al. PMID 27708353).

It is stated on that in the previous studies inhibitors showed varying trapping capabilities. It should be discussed whether the compounds actually interact with the HD domain at the region preventing NAD+ binding. Quite a range of crystal structures are available to see if there is correlation.

page 21

Please define also salt concentration used for PARP1 and PARP2.

Reviewer #3 (Remarks to the Author):

I was asked by the editor to comment on the use of HX-MS in this manuscript, thus my review will focus on technical aspects of the work and on interpretation of the results. I picked out a few other general things in my review of the manuscript.

Based on studies with BAD, a non-hydrolyzable analog of the PARP-1 substrate, the authors conclude that the folded helical domain of the PARP-1 prevents NAD⁺ from binding to the active site. However the authors are inconsistent about this claim in the manuscript, at times claiming that the helical domain “entirely prevents NAD⁺ binding” (p. 17) and in other places describing the inhibition of substrate binding as a “selective block on NAD⁺ binding” (abstract). A second finding is that in the ternary complex (PARP-1, damaged DNA, and BAD) that there are long-range changes in HX that promote higher affinity for DNA damage and also promote the unfolding of the helical domain.

CRITIQUE

The HX-MS method is described in very good detail and the experimental methodology is sound. I am confident in the reliability of the HX-MS data.

For the proposed mechanism to work, PARP-1 binding to DNA should somehow promote the unfolding of the helical domain to increase access of NAD⁺ to the binding site. Why are these data not presented? It is clear from Figure 5a, that HX of PARP-1+DNA has been measured, but it seems that it has been held back. This is a critical protein state in the model presented in Figure 6. Do the HX-MS data support the model? Does DNA binding promote unfolding in the helical domain?

Supporting figure S6 shows the effects of BAD binding in the absence of DNA at all of the on-exchange times ranging from 10¹ sec to 10⁵ sec. Only 100 and 10000 s for the ternary complex are presented in figure 5bc. No justification was provided for the selective reporting of results in Figure 5. This could simply be for the sake of data reduction. If so, why not include all of the HX-MS data in the supporting information in the same manner as figure S6?

At times, the authors claim that BAD cannot bind to PARP-1. Yet the HX data shown in figure S6, and various statements in the paper, contradict this declaration. In figure S6, the authors identified decreased protection in the PARP-1 in helix B of the helical domain induced by the presence of BAD. They also declare that this is the only effect of BAD (p. 13, lines 6-7). However, I can see evidence of subtle increases in protection in ZN2, residues 145-160, that the authors have not described. Whether the subtle increases in protection in ZN2 caused by BAD are significant requires that the HX data be presented together. With the graded color scheme it is difficult to perceive if these differences are truly significant. Plots of HX vs. t on a semilog scale with error bars would allow for a better assessment of whether these differences are actually significant. The authors should provide these results, at least for review.

Thus, based on the HX results in Figure S6, BAD does interact, at least to a limited extent, with PARP-1. In fact, the model depicted in figure 6 allows for this possibility: “infrequent BAD binding” (Fig 6, upper right corner). This may simply boil down to a matter of semantics in that it is the folded state of the

helical domain that completely blocks access to the active site, but that transient relief of the helical domain allows some limited binding that leads to basal activity.

The argument explaining altered HX by BAD only on p. 13 is somewhat unsettling. It implies slow progression towards equilibrium over a timescale of many hours between BAD and PARP-1, potentially invalidating the preceding ligand binding studies where fast equilibrium is assumed.

Figure 6 includes additional details about the binding that are not brought out clearly in the discussion. Namely that the active site "can be sub-divided [sic] into the nicotinamide binding site (N) and the adenosine binding site (a) [sic]." This idea seems to be floating around the edges of the discussion and is especially pertinent with respect to the benzamide results, but the point is never brought up explicitly.

MINOR ISSUES

The meaning of "BRCT" as one of the domains of PARP-1 was not defined in the text or captions.

p. 4: "PARP-1 interaction with DNA damage" is an odd phrase since both the nature of the damage and the nature of the interaction are ill-defined. The phrase "DNA damage" appears many times in the manuscript. Why not replace "damage" with "DNA strand breaks" to improve readability?

p. 5, last line: the reference to figure 1c should be a reference to figure 1d

p. 6, line 7: again, a reference to figure 1d

Table 2: How do the authors account for sub-stoichiometric binding well outside of experimental error?

p. 13, line 13 (and other places): the phrase "exchange event" is unclear. Does this describe a BAD binding event or a hydrogen exchange event?

p. 13 lines 7-8: The authors stated that HX was faster in α B and α F. Based on Figure S6, there are no increases in the HX rate of α F, only in α B.

p. 13, lines 22-23: "A BAD-dependent decrease in HX is also observed in other regions outside of the CAT when PARP-1 is in complex with DNA, but not in the absence of DNA (Fig. 5b, c)." This statement is not substantiated by the figures 5b, c. Instead, the effects of BAD in the absence of DNA are shown in figure S6.

p. 20: PARP-1 was exposed to benzamide during expression in E. coli. Are the authors confident that all benzamide has been removed from the active site of the purified protein?

p. 20: What data support the structures of the two inhibitors, BAD and carba-NAD⁺? Put another way, how did they validate what their provider claimed to synthesize?

The only description that I could find of the DNA used was in the NMR and fluorescence polarization sections. Is this the same DNA, with a strand break, that was used in other assays? It should be described in detail at the outset of the Methods section. In fact, I found no description of the DNA in the

main body of the manuscript. At the very least it should be described as "damaged DNA" or "strand-break DNA".

p. 24: Was the Glasoe and Long correction used for converting from pH to pD?

p. 24: How many technical replicates of HX were measured for each on-exchange time?

Figure 5a: What statistical test was used to establish significance? What was the p value? How many replicates?

Figure 5c: the blue and red coloring is difficult to discern against the grey color of the protein backbone.

Figure 6 caption uses a lowercase "(a)" instead of an uppercase "A" to refer to the adenosine binding site.

Figure S6: there are some peptides in the HX map that are not reported for both protein states. For example, there is a short, highly protected peptide near 610 in the "with BAD" data that is not in the "without BAD" data. In the main body, the authors stated that they only included HX-MS data when the peptides were recorded in both states.

Non-hydrolyzable NAD⁺ analog reveals substrate-blocking mechanism for PARP-1 and reverse allosteric communication from catalytic center to DNA binding domains

We thank each of the reviewers for their time and effort in preparing thorough and constructive reviews of our study. We appreciate the positive feedback that underscored the importance of the work. We also appreciate the comments and suggestions that have helped us to improve the manuscript. We have revised the manuscript text, figures, and supplemental material based on reviewer feedback, and we feel that we have addressed each of the concerns that were raised during the evaluation. Below, we have addressed each of the reviewer comments point-by-point, with the original reviewer comments in **bold italics**, and our responses in normal font.

Reviewers' comments:

Reviewer #1 (Remarks to the Author):

In the manuscript “Non-hydrolyzable NAD⁺ analog reveals substrate blocking mechanism for PARP-1 and reverse allosteric communication from catalytic center to DNA binding domains” Langelier and colleagues explore the structural basis of PARP-1 activation mechanism. The manuscript describes precisely for the first time how NAD is bound to the active site of PARP-1. Furthermore, it clarifies the intramolecular regulation of PARP-1 via an allosteric block by catalytic auxiliary domain HD and PARP-1 stabilisation at the DNA breaks upon NAD binding. These data will also have important implications for understanding how the clinically relevant PARP inhibitors work. The manuscript is nicely written and provides some very important new insights into the complex mechanism of PARP-1 activation.

We thank the reviewer for this positive assessment of our work.

I suggest several small changes/additions to further improve the manuscript:

DSF data:

It would be nice if the authors could provide T_m's of their reference proteins (e.g. in a supplementary table) so the reader can evaluate the dT_m values based on these reference temperatures. Also it would allow to assess the influence of the point mutation on the thermal stability of the protein.

We thank the reviewer for this suggestion and agree that it will be a good addition to the manuscript. We have added the T_M values for all the proteins tested in the study in Supplementary Table 1.

BAD dual conformers:

- i) It would be helpful to the reader if the authors would present electron density maps for the BAD dual-conformers (or at least one of them) and give estimated ratios of these conformers for each of the subunits (A-D).***

In Supplementary Figure 1, we have added an electron density map for one of the BAD dual conformers (molecule D). The approximate ratios of the conformers (occupancy values) were determined by occupancy refinement and through inspection of electron density maps and are now presented in the Results section. This information will also be contained in the coordinate file deposited in the PDB.

- ii) ***On page 7 the authors state that "the second conformation of BAD in molecule B appears to be influenced by crystal packing" and concluded that the single conformation of subunit C is the active conformation. No evidence is presented for this assumption and it would be great if the authors could elaborate on this point as the exact binding of NAD is crucial to the understanding of PARP-1's activation mechanism.***

In the text of the manuscript and the Supplementary Figure 1, we have added additional description of why we believe molecule C likely represents the active conformation based on similarity to the structures of ADP-ribosylating toxins with NAD⁺ bound.

ITC data:

- i) ***The authors demonstrate by HXMS and fluorescence polarisation that binding of BAD to full-length PARP-1 influences DNA binding and via structural re-arrangements in Zn3, BRCT, WGR and HD/ART. The authors should consider that not all of these structural changes occur at the BAD binding site. This structural changes will, therefore, contribute to the measured heat in the ITC experiment, thus contributing to the calculated Kd value, without being a direct consequence of contacts established between ligand and protein. The authors should comment this.***

Based on the general agreement in BAD binding affinities across the different ITC experiments, we are confident that the measured heat is primarily reporting on the ligand-protein interaction; however, we agree with reviewer that the ITC experiments with PARP-1/DNA represents a complex system that could potentially have additional contributions to the measured heat besides just ligand binding. We have thus added text to the Results section to comment on this aspect of the ITC data.

- ii) ***Figure 4e: One data point is at a molar ration <0. It would be good if the authors could double check their data.***

Thank you for identifying this error. It has been corrected.

Page 5 last line – Figure reference should be to Fig. 1d.

Now corrected – thank you.

It would be nice if the authors could discuss whether similar mechanisms to one described here can be found in other proteins.

There are some similarities to the NAD⁺ hydrolyzing toxins that complex with an anti-toxin to prevent NAD⁺ depletion in the host system. However, in these systems that anti-toxin appears to impose major changes on the NAD⁺ binding site conformation, whereas the PARP-1 substrate-blocking mechanism of the HD identified in our study does not influence the NAD⁺ binding site conformation, but rather presents a steric block that can be more easily regulated, consistent with PARP-1 function in multiple cellular contexts. We have added a brief portion of text to the Discussion to highlight these similarities and differences.

Reviewer #2 (Remarks to the Author):

Reviewer comments

The manuscript by Langelier et al. reports a crystal structure of PARP-1 catalytic domain with a substrate analog and describe a two-way communication between the catalytic C-terminal ADP-ribosylating domain and the N-terminal DNA binding domains. Despite significant efforts by multiple research teams there are no substrate bound structures for the enzymes of the PARP family and the structure reported here is the closest one to the actual NAD⁺ substrate. The study itself is adding on to previous work published by the authors where they have studied activation mechanism of the PARP1 enzyme by binding to DNA. The manuscript provides further evidence for the allostery, but will also explain earlier observations through detailed analysis of the protein. The manuscript is fluent and well written and will increase our understanding on how the key enzyme in DNA damage recognition works.

Thank you for this positive feedback on our study.

My comments to the author are listed below:

Page 5 and Figure 1c

The authors have used chicken PARP1 in structural comparison and also describe binding of partial carba-NAD⁺ to this particular structure. This carba-NAD⁺ fragment has been thought to mark the acceptor site as mentioned and it should be added to the figure. It is not entirely clear why chicken structure was selected as there are crystal structures of the human enzyme also available. There are residues differences (E/Q763) and this should be explained in the figure legend.

We have historically used the chicken PARP-1 structure as a reference as it provides unique information relative to other structures, due to the partial carba-NAD⁺ molecule captured in the chicken structure. However, we agree that this aspect of the chicken PARP-1 structure is not relevant to the main point of Figure 1c, so we have now altered the figure to include a human PARP-1 structure as a reference instead, which still illustrates the same point but avoids confusion (e.g. there are no longer residue differences to explain).

Authors should explain, why in Fig. 1a (monomer C) the active site loop is not visible, while it is present in the other monomers having the same BAD binding mode (Figure S1). Despite this the binding mode in monomer C would be the catalytically relevant mode.

The active site loop of PARP-1 is normally involved in contacts with the HD. The deletion of the HD removes these contacts, and appears to provide a level of flexibility to the active site loop in the absence of the HD. For monomers A, B, and D, the active site loop is located in crystal packing environments that are likely to help stabilize the active site loop, whereas the active site loop of monomer C is more exposed to solvent and lacks nearby contacts. The lack of local contacts for the active site loop in monomer C has likely led to a greater level of flexibility. We observe electron density that accounts for the location of the active site loop in monomer C; however, the density is only apparent when the electron density map is contoured at a lower sigma level (~0.5 sigma). We did not feel that we could confidently model this region, but it is clear from the location of the observable density that it will occupy the same general position as seen in monomers A, B, and D. We have added a brief discussion of the active site loop to the Results section.

Are there any interactions of BAD and the D-loop/ASL and or is it just disordered upon binding. In the crystal structure used as a model in molecular replacement the ASL is visible. Are there any

changes observed between the structures? This could be added as a supplementary figure.

Tyrosine 889 is the only residue considered to be part of the active site loop that forms an interaction with BAD (shown in Figure 2b, and Supplemental Figure 1). Tyr 889 is modeled in each of the PARP-1 monomers, including monomer C. As mentioned in the response to the last point, we believe that any change in the flexibility or ordered-ness of the active site loop is a consequence of the HD deletion and/or the presence/absence of crystal contacts. Based on the analysis of the BAD/PARP-1 structure and comparison with other PARP-1 structures (including the model used in molecular replacement), we do not believe that the active site loop necessarily becomes disordered upon binding BAD, and it does not appear to have a major role in BAD binding based on previous mutagenesis of residues on this loop, which we now make clear and reference in the revised manuscript.

Now the authors have solved a co-crystal structure of close substrate analog yet they do not discuss the catalytic mechanism at all, which is unsatisfying in the structural analysis. Is the carba-NAD⁺ fragment at all in a suitable position for chain elongation?

The carba-NAD⁺ molecule from the chicken PARP-1 structure is in a reasonable position to support the ADP-ribose chain elongation mechanism that has been proposed. However, we feel that the exact positioning will be best determined with a structure of human PARP-1 with a molecule bound to the acceptor site, which will alleviate concerns that minor human/chicken sequence differences have influenced the positioning of the acceptor ADP-ribose molecule. Since we are not yet in a position to provide a clear advance in the chain elongation mechanism, we have decided to simply present our structure and make reference to the published chicken PARP-1 structure. However, we have now included an image of the alignment of chicken PARP-1/carba-NAD⁺ with our PARP-1/BAD structure in Supplementary Figure 1.

“engineered link” is included in the Figure 1 a, but it is not defined in the legend or text and the rationale for the engineering is not given in the methods. This should be clarified.

The linker serves to replace the HD domain that is deleted, providing enough residues to connect alpha helix A to alpha helix G (shorter linkers of 6 and 7 residues were not effective in bridging the distance). This work was previously described and we have made this reference clear in the revised manuscript.

Please define in figure legend the “DNA”. This is different for the enzymes and would help the reader to understand the experiment when looking at the figures.

A description of the DNA used in the various experiments has been added to the relevant Figure legends.

page 6

IC50 curves in Figure 1f should be fitted. How come the IC50 is so much higher than the Kd measured with ITC?

We have now fitted the IC50 curves, which are an average of three independent experiments, in Figure 1f. The values obtained (8.6 μ M for BAD; 4.6 μ M for benzamide) are similar to the ones obtained using ITC (2 to 6 μ M).

page 7

Before stating that there is a single conformation in molecule C it should be stated that there are multiple conformations for the BAD. Now it is difficult to grasp when reading the first time what is

meant.

We have now reorganized this section of the text to address this issue. We have stated first that there are multiple BAD conformations, as suggested.

page 9

Authors refer to an old publication, by Rippmann et al. which measured a K_m of TNKS1 to be 1.5 mM. In a recent article Thorsell et al. (PMID: 28001384) measured a K_m for TNKS1 to be very similar to that of PARP1. Therefore this rationale may not be valid. One can not effectively measure affinities of a compound to different proteins using DSF. Could it be that the ΔT_m for TNKS1 is just overall smaller when substrate binds? It appears as if a concentration series was measured and then it would be possible to fit this and get an estimated K_d (Alternatively ITC could be used)

Similar to Rippmann et al., we have observed in TNKS1 catalytic activity assays that the apparent K_m for NAD^+ is in the range of 1.5 mM. Using DSF, the observed ΔT_m for TNKS1 increased as we increased the BAD concentration to above 1 mM. The purpose of our presented TNKS1 data is primarily to demonstrate that BAD is free to bind to TNKS1, since it does not contain an inhibitory HD, and to demonstrate that BAD will serve as a useful chemical tool for TNKS1 analysis. The actual K_m of TNKS1 is not a focus of the current study, and we stated the increased concentration of BAD simply to clarify why we used a higher concentration for TNKS1 than for PARP-1.

Authors should also list the actual T_m values for the proteins at least in the supplementary file and not just the delta.

As noted in our response to the same request from reviewer 1, we have added the T_m values for all of the proteins tested in the study in Supplementary Table 1.

Text mentions 4 mM BAD and figure legend 1.25 mM.

This error has been corrected in the Figure legend – thank you.

Define the errors in Table 2.

This information was located in the Methods, but it has now also been added to Table 2 so that it can be viewed alongside the data.

page 16

Comparison of the NAD^+ binding to toxins and to PARP1 should be shown.

We have expanded Supplementary Figure 1 to include a comparison of NAD^+ binding to ADP-ribosyl transferases and BAD binding to PARP-1.

page 19

There is also a study showing potential contribution of the CAT to the DNA binding (Obaji et al. PMID 27708353).

We have added this reference to the text.

It is stated on that in the previous studies inhibitors showed varying trapping capabilities. It should be

discussed whether the compounds actually interact with the HD domain at the region preventing NAD⁺ binding. Quite a range of crystal structures are available to see if there is correlation.

We agree that this is an interesting and important area moving forward, but we feel it would be too speculative to discuss possibilities at this point. Our visual inspection of the currently available structures of PARP-1 CAT domain bound to various inhibitors has not yielded a clear connection between HD contacts and trapping potential. We feel that a discussion of these differences will be more informative in the context of experimental data that assesses inhibitor contributions to PARP-1 dynamics, and mutagenesis that targets potential inhibitor-specific contacts with the HD.

page 21

Please define also salt concentration used for PARP1 and PARP2.

The salt concentration used in the DSF experiment for PARP-1 and PARP-2 has been added.

Reviewer #3 (Remarks to the Author):

I was asked by the editor to comment on the use of HX-MS in this manuscript, thus my review will focus on technical aspects of the work and on interpretation of the results. I picked out a few other general things in my review of the manuscript.

Based on studies with BAD, a non-hydrolyzable analog of the PARP-1 substrate, the authors conclude that the folded helical domain of the PARP-1 prevents NAD⁺ from binding to the active site. However the authors are inconsistent about this claim in the manuscript, at times claiming that the helical domain “entirely prevents NAD⁺ binding” (p. 17) and in other places describing the inhibition of substrate binding as a “selective block on NAD⁺ binding” (abstract). A second finding is that in the ternary complex (PARP-1, damaged DNA, and BAD) that there are long-range changes in HX that promote higher affinity for DNA damage and also promote the unfolding of the helical domain.

CRITIQUE

The HX-MS method is described in very good detail and the experimental methodology is sound. I am confident in the reliability of the HX-MS data.

We thank the reviewer for the positive feedback.

For the proposed mechanism to work, PARP-1 binding to DNA should somehow promote the unfolding of the helical domain to increase access of NAD⁺ to the binding site. Why are these data not presented?

We decided to not highlight this data in the current manuscript, since it was the focus of our previous study (Dawicki-McKenna, et al. 2015 Mol Cell). We thank the reviewer for alerting us that our previously reported findings were not well integrated into the presentation of the new data. To address this issue, we have added additional references to our previous work, we have pointed out in the text that the HD unfolding can be observed in Supplementary Figure 6 and 7, and we have modified Fig. 5b to indicate the portions of the HD that unfold when PARP-1 binds to DNA, based on our previous work.

It is clear from Figure 5a, that HX of PARP-1+DNA has been measured, but it seems that it has been

held back. This is a critical protein state in the model presented in Figure 6. Do the HX-MS data support the model? Does DNA binding promote unfolding in the helical domain?

Yes, the HX-MS data fully support the model that DNA binding promotes unfolding of the HD, as described in Dawicki-McKenna et al. 2015. We have modified the text of the manuscript to make this point more clearly. We apologize for the confusion.

Supporting figure S6 shows the effects of BAD binding in the absence of DNA at all of the on-exchange times ranging from 10¹ sec to 10⁵ sec. Only 100 and 10000 s for the ternary complex are presented in figure 5bc. No justification was provided for the selective reporting of results in Figure 5. This could simply be for the sake of data reduction. If so, why not include all of the HX-MS data in the supporting information in the same manner as figure S6?

We thank the reviewer for helping us to clarify our rationale for presenting the data in Figure 5 versus the supplement. We now explain in the main text our reason for focusing on specific timepoints. In short, we first recorded MS spectra at 5 different timepoints (the percent HX is all shown in the supplement, as the reviewer points out) and analyzed the HX. The 100 sec timepoint revealed substantial changes in regions that are either unfolded or rapidly sampling unfolded states (such as the HD domain that is a focus of our paper and other loop regions) because at later timepoints they are essentially completely exchanged. On the other hand, the 10000 sec timepoint revealed substantial changes in well-folded regions (such as the CAT domain, and other peptides in folded regions), because there was only minimal exchange at earlier timepoints in all regions. In this way, the structural modeling and other data presentation in the main figure set is focused on the most easily interpreted findings from our HXMS experiments, while the supplement can highlight the totality of the peptide data that guided us from the beginning to the end of our analysis. Our revised paper is stronger because we have now explained this rationale in clearer terms. As part of this improvement, in order to be clear for the general reader, we have also included raw mass spectra for all timepoints for some key peptides (Supplementary Figure 8), so the experiment is as transparent as possible for anyone to assess.

At times, the authors claim that BAD cannot bind to PARP-1. Yet the HX data shown in figure S6, and various statements in the paper, contradict this declaration. In figure S6, the authors identified decreased protection in the PARP-1 in helix B of the helical domain induced by the presence of BAD. They also declare that this is the only effect of BAD (p. 13, lines 6-7). However, I can see evidence of subtle increases in protection in ZN2, residues 145-160, that the authors have not described. Whether the subtle increases in protection in ZN2 caused by BAD are significant requires that the HX data be presented together. With the graded color scheme it is difficult to perceive if these differences are truly significant. Plots of HX vs. t on a semilog scale with error bars would allow for a better assessment of whether these differences are actually significant. The authors should provide these results, at least for review.

Thus, based on the HX results in Figure S6, BAD does interact, at least to a limited extent, with PARP-1. In fact, the model depicted in figure 6 allows for this possibility: “infrequent BAD binding” (Fig 6, upper right corner). This may simply boil down to a matter of semantics in that it is the folded state of the helical domain that completely blocks access to the active site, but that transient relief of the helical domain allows some limited binding that leads to basal activity.

Yes, the reviewer’s description is precisely what we hoped to convey with our model, that the folded helical domain completely blocks access, and that transient relief of the helical domain structure allows some limited binding, which is necessary to explain basal activity. We have revised the manuscript to

make sure that our model is more clearly presented, and that we are careful in stating that the folded state of the HD blocks BAD binding.

Regarding the HXMS analysis of BAD binding to PARP-1 in the absence of DNA, we observed no evidence of robust binding to BAD, which is indeed consistent with the ITC and DSF analysis and supports the model mentioned above. We originally also placed some focus on subtle differences we noted in the same HD region that we knew to be affected by BAD interaction with PARP-1 in the presence of DNA, and interpreted this as evidence of infrequent binding that is expected to support basal activity. As the reviewer very helpfully points out, our original presentation probably over-simplified the HXMS analysis of BAD binding to PARP-1 in the absence of DNA, since there are also subtle changes that occur in other regions that we cannot explain through a BAD binding event. In the revised manuscript, we have decided to not focus on the HXMS analysis of low level, transient binding of BAD in the absence of DNA, since this is a minor component of the study. Instead, the HXMS analysis can now remain focused on what we think are the more easily interpretable aspects (e.g. major messages of the paper: the inability of BAD to robustly bind to PARP-1 in the absence of DNA damage, and the allosteric changes transmitted from the NAD⁺ binding site through PARP-1 when it is bound by BAD). As part of the revisions, we have added a supplementary figure (new Supplementary Figure 8), where we highlight the raw MS data for key peptides from the primary regions of interest (such as the HD and CAT). With all of these improvements/additions, we think this presentation in our revised manuscript is much more effective, and we thank the reviewer for helping us improve this part of the paper.

The argument explaining altered HX by BAD only on p. 13 is somewhat unsettling. It implies slow progression towards equilibrium over a timescale of many hours between BAD and PARP-1, potentially invalidating the preceding ligand binding studies where fast equilibrium is assumed.

As mentioned in a previous response, we are no longer focusing on the HXMS analysis of BAD binding to PARP-1 in the absence of DNA, so the argument referenced on p. 13 is no longer a part of the text. It is still worth noting that we do not think that the argument supported a slow progression toward equilibrium, even if our description indeed gave this impression. Notably, the NMR data was acquired over the time scale of hours (incubation time plus data acquisition time), yet did not show evidence of PARP-1 binding to BAD unless DNA was present. We feel that our data presentation is now consistent with the final model, as covered in the preceding response.

Figure 6 includes additional details about the binding that are not brought out clearly in the discussion. Namely that the active site "can be sub-divided [sic] into the nicotinamide binding site (N) and the adenosine binding site (a) [sic]." This idea seems to be floating around the edges of the discussion and is especially pertinent with respect to the benzamide results, but the point is never brought up explicitly.

We now have added a better description of Figure 6 in the discussion to cover this information.

MINOR ISSUES

The meaning of "BRCT" as one of the domains of PARP-1 was not defined in the text or captions.

We have added the description of the BRCT acronym.

p. 4: "PARP-1 interaction with DNA damage" is an odd phrase since both the nature of the damage and the nature of the interaction are ill-defined. The phrase "DNA damage" appears many times in

the manuscript. Why not replace “damage” with “DNA strand breaks” to improve readability?

We have replaced DNA damage with DNA strand breaks throughout the text where appropriate to address this concern.

p. 5, last line: the reference to figure 1c should be a reference to figure 1d

Corrected – thank you.

p. 6, line 7: again, a reference to figure 1d

Corrected – thank you again.

Table 2: How do the authors account for sub-stoichiometric binding well outside of experimental error?

We anticipate that either the active fraction of the catalytic domain is not 100%, or the concentration of the catalytic domain constructs is not entirely accurate. The analysis of full-length PARP-1 yielded the expected stoichiometry and we are therefore not concerned that this could affect the conclusions related to the ITC experiments. We have added text in the Methods section to acknowledge this aspect of the ITC data.

p. 13, line 13 (and other places): the phrase “exchange event” is unclear. Does this describe a BAD binding event or a hydrogen exchange event?

We have used this phrase to describe a hydrogen exchange event. We have worked to make this more evident in the revised text.

p. 13 lines 7-8: The authors stated that HX was faster in αB and αF . Based on Figure S6, there are no increases in the HX rate of αF , only in αB .

Increases in the HX of peptides within the αF helix can be observed in Supplementary Figure 6. For instance, if you examine the peptide 771-778 from αF , you'll see that at the last timepoint (10^5 sec), the peptide is more deuterated in the presence of BAD than in its absence.

Legend: Ribbon plot for a 771-778 peptide from αF helix in the absence of DNA with or without BAD. The darker, more orange color of the last bar in the presence of BAD compared to the peptide in the absence of BAD indicates higher deuterium level of this peptide at 10^5 sec.

p. 13, lines 22-23: “A BAD-dependent decrease in HX is also observed in other regions outside of the CAT when PARP-1 is in complex with DNA, but not in the absence of DNA (Fig. 5b, c).” This statement is not substantiated by the figures 5b, c. Instead, the effects of BAD in the absence of DNA

are shown in figure S6.

We have added a reference to Supplementary Figure 6 at the end of the sentence.

p. 20: PARP-1 was exposed to benzamide during expression in E. coli. Are the authors confident that all benzamide has been removed from the active site of the purified protein?

We are confident that the benzamide has been effectively removed. PARP-1 is purified using three chromatographic steps with extensive washing volumes using solutions that have no benzamide present. Since PARP-1 affinity for benzamide is in the micromolar range, we are confident that the benzamide has been fully removed from the purified sample. Moreover, we have observed no differences in full-length PARP-1 catalytic activity assays using protein grown with or without benzamide.

p. 20: What data support the structures of the two inhibitors, BAD and carba-NAD⁺? Put another way, how did they validate what their provider claimed to synthesize?

The company that we used to synthesize the compounds provided a certificate of analysis for both compounds, which included NMR spectra and LCMS that are consistent with the expected structures. Additionally, our crystal structure and NMR experiments are consistent with the expected structure of BAD. For carba-NAD⁺, we also collected our own NMR spectra that confirm that the compound has the expected structure. We have added more text to the Methods section to reflect our confidence in the reagents used.

The only description that I could find of the DNA used was in the NMR and fluorescence polarization sections. Is this the same DNA, with a strand break, that was used in other assays? It should be described in detail at the outset of the Methods section. In fact, I found no description of the DNA in the main body of the manuscript. At the very least it should be described as "damaged DNA" or "strand-break DNA".

We have added a description of the DNA used in each assay in the Figure legends where relevant.

p. 24: Was the Glasoe and Long correction used for converting from pH to pD?

Yes, it was. Please see the addition on pg. 25 in the Methods section.

p. 24: How many technical replicates of HX were measured for each on-exchange time?

We clarified this point in the Methods section. In short, and as outlined in a previous response above, HX was done in triplicate for 5 timepoints. After we completed analysis from each timepoint, we focused our interest on 100 sec and 10000 sec timepoints. As stated in the figure legend, as well, the values shown in Fig. 5a represent the mean of three independent measurements with error bars calculated as a standard deviation of these three measurements.

Figure 5a: What statistical test was used to establish significance? What was the p value? How many replicates?

We clarified this in the legend of Fig. 5. Values were reported as an average of three independent measurements with standard deviations used for error bars. p value between measurements indicated with * were under 0.0005. p value for a single difference indicated with ns was above 0.5.

Figure 5c: the blue and red coloring is difficult to discern against the grey color of the protein backbone.

We altered the coloring and presentation of Figure 5c to address this issue.

Figure 6 caption uses a lowercase “(a)” instead of an uppercase “A” to refer to the adenosine binding site.

We have corrected this mistake.

Figure S6: there are some peptides in the HX map that are not reported for both protein states. For example, there is a short, highly protected peptide near 610 in the “with BAD” data that is not in the “without BAD” data. In the main body, the authors stated that they only included HX-MS data when the peptides were recorded in both states.

We can see why our original version of the paper was confusing on this point. The term used in the Methods section "both conditions" referred only to the difference plots shown in Figure 5a, since the peptide has to be present in both experiments to calculate the difference in deuteration. In Supplementary Figure 6, however, we report actual deuteration levels, not the differences, thus we did not exclude peptides that were present in one dataset, but not the other. We modified the description in the Methods section "HXMS plotting" to clarify this, and we thank the reviewer for helping us improve this aspect of data presentation.